# WEIGHTED RISK INVARIANCE FOR DENSITY-AWARE DOMAIN GENERALIZATION

## ABSTRACT

Learning how to generalize training performance to unseen test distributions is essential to building robust, practically useful models. To this end, many recent studies focus on learning invariant features from multiple domains. However, we observe that the performance of existing invariant learning methods can degrade under covariate shift. To address this problem, we focus on finding invariant predictors from multiple, potentially shifted invariant feature distributions. We propose a novel optimization problem, Weighted Risk Invariance (WRI), and we show that the solution to this problem provably achieves out-of-distribution generalization. We also introduce an algorithm to practically solve the WRI problem that learns the density of invariant features and model parameters simultaneously, and we demonstrate our approach outperforms previous invariant learning methods under covariate shift in the invariant features. Finally, we show that the learned density over invariant features effectively detects when the features are out-of-distribution.

## 1 INTRODUCTION

Although traditional machine learning methods can be incredibly effective, many are based on the *i.i.d. assumption* that the training and test data are independent and identically distributed. As a result, these methods are often brittle to distribution shift, failing to generalize training performance to *out-of-distribution* (OOD) data (Torralba & Efros, 2011; Beery et al., 2018; Hendrycks & Dietterich, 2019; Geirhos et al., 2020). Distribution shifts abound in the real world: as general examples, we encounter them when we collect test data under different conditions than we collect the training data (Adini et al., 1997; Huang et al., 2006), or when we train on synthetic data and evaluate on real data (Chen et al., 2020; Beery et al., 2020). Learning to generalize under distribution shift, then, is a critical step toward practical deployment.

To achieve generalization, a typical setup is to use multiple training environments with the hope that, if the model learns what is invariant across the training environments, it can leverage that invariance on an unseen test environment as well. An extensive line of research therefore focuses on learning statistical relationships that are invariant across training environments (Ganin et al., 2016; Tzeng et al., 2017; Li et al., 2021). Of these, recent works focus on learning an invariant relationship between the data $X$ and label $Y$ distributions, as such relationships are posited to result from invariant causal mechanisms (Pearl, 1995; Schölkopf et al., 2012; Peters et al., 2016; Rojas-Carulla et al., 2018). Indeed, several works have found that learning features $\Phi(X)$ such that the label conditional distribution $p(Y|\Phi(X))$ is invariant across environments is an effective method of uncovering the coefficients of a data-generating causal model (Arjovsky et al., 2019; Krueger et al., 2021; Wald et al., 2021; Eastwood et al., 2022).

Compared to predictors that have strict distribution shift assumptions (e.g. covariate shift (Muandet et al., 2013; Robey et al., 2021), label shift (Lipton et al., 2018; Liu et al., 2021c), etc.), predictors that rely on conditionally invariant features can be seen as more well-motivated and more generally robust (Koyama & Yamaguchi, 2020). However, we demonstrate that existing methods for finding these predictors can struggle or even fail when the conditionally invariant features undergo covariate shift. We will see that covariate shift introduces two types of challenges: first, it could adversely impact sample complexity Shimodaira (2000), and second, it could lead some invariance-enforcing penalties to incorrectly identify the conditionally invariant features.

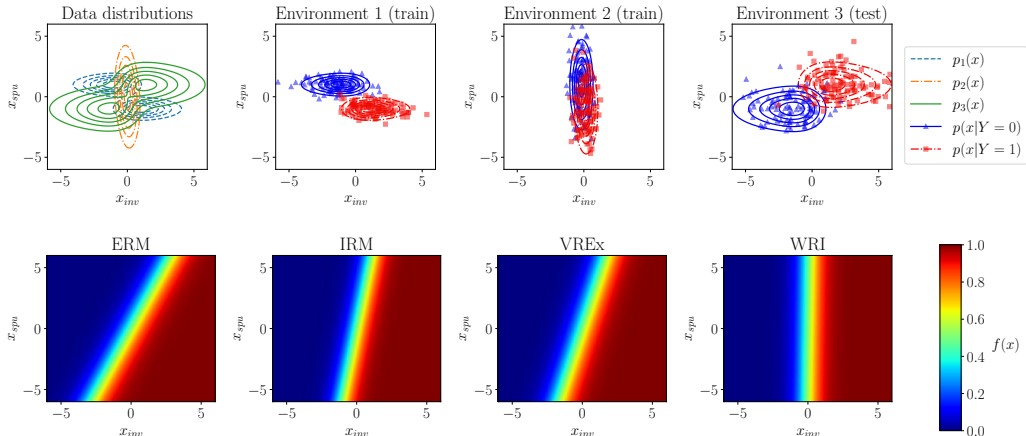

**Figure 1:** Setup (top) and trained predictors (bottom) for our linear causal model, following Assumption 1, when the marginal distributions of the invariant features differ. In this setup, the solution to WRI is a nearly invariant predictor. Full parameters for this simulation can be found in Appendix C.

In this work, we argue that we can mitigate the effects of covariate shift by accounting for the underlying structure of the data in our learning process. We therefore introduce a novel optimization problem, Weighted Risk Invariance (WRI), that uses information from the feature distributions to adjust for shifts. Figure 1 shows that solving for WRI recovers a nearly invariant predictor, even in the difficult case of heteroskedasticity and covariate shift. We theoretically show that solutions to WRI achieve OOD generalization by learning to reject spurious correlations, and we provide an algorithm to solve WRI in practice. Finally, we show that under distribution shift in the invariant features, the WRI solution outperforms state-of-the-art baselines.

In summary, our main contributions are:

- We introduce a novel invariant learning method that utilizes Weighted Risk Invariance (WRI) to tackle covariate shift in the invariant features. We theoretically show that the solution of WRI recovers an invariant predictor under a linear causal setting (Prop. 1, Thm. 2).

- We propose an algorithm to empirically solve WRI which involves learning the model parameters and the density of the invariant feature distribution using alternating minimization (§3.2).

- We verify the efficacy of our approach with experiments on simulated and real-world data. We find that, under covariate shift in the invariant features, our method outperforms state-of-the-art baselines on benchmarks like ColoredMNIST and DomainBed. Moreover, our learned invariant feature density reports when invariant features are outside of the training distribution, making it useful for downstream tasks like OOD detection (§4).

## 2 PRELIMINARIES

### 2.1 DOMAIN GENERALIZATION

Domain generalization was first posed (Blanchard et al., 2011; Muandet et al., 2013) as the problem of learning some invariant relationship from multiple training domains/environments $\mathcal{E}_{tr} = \{e_1, \ldots, e_k\}$ that we assume to hold across the set of all possible environments we may encounter $\mathcal{E}$. Each training environment $E = e$ consists of a dataset $D^e = \{(\mathbf{x}_i^e, y_i^e)\}_{i=1}^{n_e}$, and we assume that data pairs $(\mathbf{x}_i^e, y_i^e)$ are sampled i.i.d from distributions $p_e(X^e, Y^e)$ with $\mathbf{x} \in \mathcal{X}$ and $y \in \mathcal{Y}$. (When the context is clear, we write the data-generating random vector $X$, output random variable $Y$, and their realizations without the environment superscript.)

For loss function $\ell$, we define the statistical *risk* of a predictor $f$ over an environment $e$ as

$$\mathcal{R}^e(f) = \mathbb{E}_{p_e(X^e, Y^e)} \left[ \ell(f(\mathbf{x}^e), y^e) \right], \tag{1}$$

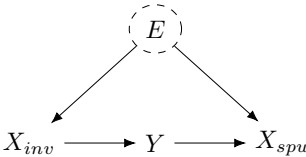

**Figure 2:** Causal graph depicting our data-generating process. The environment $E$ is dashed to emphasize it takes on unobserved values at test time.

and its empirical realization as

$$\frac{1}{|D^e|} \sum_{i=1}^{|D^e|} \ell(f(\mathbf{x}_i^e), y_i^e). \tag{2}$$

Empirical Risk Minimization (ERM) (Vapnik, 1991) aims to minimize the average loss over all of the training samples

$$\mathcal{R}_{ERM}(f) = \frac{1}{|\mathcal{E}_{tr}|} \sum_{e \in \mathcal{E}_{tr}} \mathcal{R}^e(f). \tag{3}$$

Unfortunately, ERM fails to capture distribution shifts across training environments (Arjovsky et al., 2019; Krueger et al., 2021), and so can fail catastrophically depending on how those distribution shifts extend to the test data. (Previous works point to cases where ERM successfully generalizes, but the reason is not well understood (Vedantam et al., 2021; Gulrajani & Lopez-Paz, 2020), and its success does not hold across all types of distribution shifts (Ye et al., 2022; Wiles et al., 2021).) In our work, we therefore aim to learn a predictor that captures the underlying causal mechanisms instead, under the assumption that such mechanisms are invariant across the training and test environments.

## 2.2 OUR CAUSAL MODEL

We assume that input $X$ contains both causal/invariant and spurious components such that there exists a deterministic mechanism to predict the label $Y$ from invariant components $X_{inv}$, while the relationship between spurious components $X_{spu}$ and label $Y$ depends on the environment. More explicitly, we say that $Y \not\perp\!\!\!\perp E \mid X_{spu}$. The observed features $X$ are a function of the invariant and spurious components, or $X = g(X_{inv}, X_{spu})$, which we assume is injective so that there exists $g^\dagger(\cdot)$ that recovers $X_{spu}, X_{inv}$ almost surely. We represent this data-generating process with the causal graph in Figure 2. From the graph, we observe that $p_e(Y \mid X_{inv}) = p_{\tilde{e}}(Y \mid X_{inv})$ for any $e, \tilde{e} \in \mathcal{E}$, i.e. it is fixed across environments. On the other hand, the distribution $p_e(Y \mid X_{spu})$ is not fixed across various $e \in \mathcal{E}$ and we assume it may change arbitrarily. We summarize these assumptions below.

**Assumption 1** (Our problem setting). Following Figure 2, observed data for environment $E = e$ is generated from random vectors $X^e \in \mathbb{R}^d$ that are composed of invariant components $X_{inv}^e \in \mathbb{R}^{d_{inv}}$ and spurious components $X_{spu}^e \in \mathbb{R}^{d_{spu}}$. The conditional distribution $p_e(Y^e|X_{inv}^e)$ is equal for all environments in our training set $e \in \mathcal{E}_{tr}$, while the distributions $p_e(X_{spu}^e|Y^e)$ are not equal for all training environments $e \in \mathcal{E}_{tr}$.

In order to learn a predictor that depends on causal mechanisms, we aim to search among predictors on $X_{inv}$ with output conditional distributions that are equal for all training environments. We would like this equality to extend beyond the training environments as well, so we define an invariant predictor to formalize the qualities we are looking for.

**Definition 1.** We say that a predictor $f : \mathcal{X} \to \mathcal{Y}$ is *spurious-free* over a set of environments $\mathcal{E}'$ if $f(X) = f(X_{inv})$ for all environments $e \in \mathcal{E}'$. The spurious-free predictor that performs best on all environments is the *optimally invariant* predictor, which we call an *invariant predictor* for short.

Invariant predictors, or representations, are usually defined as models that satisfy $p_e(Y \mid f(X)) = p_{e'}(Y \mid f(X))$ for any $e, e' \in \mathcal{E}'$ (Arjovsky et al., 2019). For the settings we study in this paper, the optimal invariant predictor as defined in Definition 1 satisfies this property.

## 2.3 INVARIANT LEARNING UNDER COVARIATE SHIFT

Causal predictors assume there exist features $X_{inv} = \Phi(X)$ such that the mechanism that generates the target variables $Y$ from $\Phi(X)$ is invariant under distribution shift; this leads to the conditional distribution $p(Y|\Phi(X))$ being invariant across environments (Schölkopf et al., 2012; Peters et al., 2016). Conditionally invariant features induce other forms of invariance, and recent works search for these proxy forms of invariance in an attempt to recover conditionally invariant features from complex, high-dimensional data. Popular methods along this line include Invariant Risk Minimization (IRM) (Arjovsky et al., 2019) and Risk Extrapolation (REx) (Krueger et al., 2021). For simplicity, we call conditionally invariant features *invariant* when the context is clear, using terminology consistent with previous works (Liu et al., 2021a; Ahuja et al., 2021).

## 3 DENSITY-AWARE GENERALIZATION

### 3.1 DENSITY-WEIGHTED RISK INVARIANCE

Risk invariance is a widely used signature of prediction on invariant features. Unfortunately, heteroskedasticity can obscure the relationship between the invariant features and the label so that under invariant feature covariate shift, risk invariance no longer holds. In order to compare risks across environments, we need a reweighting that respects the underlying invariant feature distribution. To this end, we introduce the idea of weighted risk, an asymmetric measure of risk over one environment where we weigh by the invariant feature density of another environment. We define this concept formally below.

**Definition 2.** We define the weighted risk between two environments $e_i$ and $e_j$ as

$$\mathcal{R}^{e_i,e_j}(f) = \mathbb{E}_{p_{e_i}(X^{e_i},Y^{e_i})}\left[\ell(f(\mathbf{x}),y) \cdot p_{e_j}(\mathbf{x}_{inv})\right]. \tag{4}$$

Weighted risk invariance between two environments means that $\mathcal{R}^{e_i,e_j}(f) = \mathcal{R}^{e_j,e_i}(f)$, and weighted risk invariance over a set of more than two environments means that weighted risk invariance holds for all pairwise combinations of environments in the set.

We now present our first result, that a spurious-free predictor leads to weighted risk invariance in a general causal setting.

**Proposition 1.** *Let Assumption 1 hold over a set of environments $\mathcal{E}'$. If a predictor $f$ is spurious-free over $\mathcal{E}'$, then weighted risk invariance holds over $\mathcal{E}'$.*

This result shows that weighted risk invariance is a signature of spurious-free prediction, and it is easy to verify that the optimal invariant predictor also satisfies weighted risk invariance. However, to properly demonstrate how weighted risk invariance could be used to train a *generalizable* predictor, we need to show that the result holds in the opposite direction, over a larger set of environments. Thus, as in previous work on principled methods for learning invariant models (Arjovsky et al., 2019; Krueger et al., 2021; Wald et al., 2021), we also prove that predictors with weighted risk invariance discard spurious features $X_{spu}$ for OOD generalization under general position conditions (see Appendix A). We show this for the case of linear regression, where we have the following data-generating process for training environment $i \in [k]$:

$$
\begin{aligned}
Y &= \mathbf{w}_{inv}^* \mathbf{x}_{inv} + \varepsilon, \ \varepsilon \sim \mathcal{N}(0, \sigma_y^2) \\
X_{spu} &= \mu_i y + \eta, \ \eta \sim \mathcal{N}(0, \Sigma_i).
\end{aligned}
\tag{5}
$$

**Theorem 2.** *Consider a regression problem following the data generating process of equation 5. Let $\mathcal{E}'$ be a set of environments that satisfy general position. A linear regression model $f(\mathbf{x}) = \mathbf{w}^\mathsf{T} \mathbf{x}$ with $\mathbf{w} = [\mathbf{w}_{inv}, \mathbf{w}_{spu}]$ that satisfies weighted risk invariance w.r.t the squared loss must satisfy $\mathbf{w}_{spu} = 0$.*

Our result shows that solving for weighted risk invariance allows us to learn a predictor that discards spurious correlations, letting performance generalize across environments. In order to learn an invariant predictor, we therefore propose to learn a predictor $f_{WRI}$ that combines generalizability with good average performance:

$$f_{WRI} = \arg\min_f \sum_{e \in \mathcal{E}_{tr}} \mathcal{R}^e(f) + \lambda \sum_{\substack{e_i \neq e_j, \\ e_i,e_j \in \mathcal{E}_{tr}}} (\mathcal{R}^{e_i,e_j}(f) - \mathcal{R}^{e_j,e_i}(f))^2. \tag{6}$$

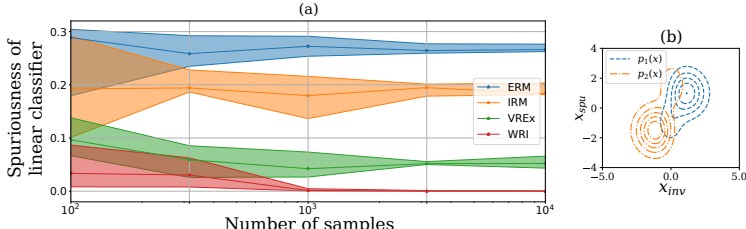

**Figure 3:** (a) We demonstrate the finite sample behavior of ERM, IRM, VREx, and WRI methods in the case where the distribution shift is large. We sample data from the distributions shown in (b). Even as the number of samples increases, ERM, IRM and VREx methods continue to select spurious classifiers while the WRI method quickly converges to an invariant classifier.

The first term enforces ERM and the second term enforces weighted risk invariance, and hyperparameter $\lambda$ controls the weight of the WRI regularization.

**Comparison to REx**   Risk invariance across environments is a popular and practically effective method for achieving invariant prediction (Xie et al., 2020; Krueger et al., 2021; Liu et al., 2021b; Eastwood et al., 2022) that was formalized by REx Krueger et al. (2021). Specifically, the VREx objective is a widely used approach of achieving risk invariance, where the variance between risks is penalized in order to enforce equality of risks:

$$\mathcal{R}_{VREx}(f) = \mathcal{R}_{ERM}(f) + \lambda_{VREx}\text{Var}(\{\mathcal{R}^{e_1}(f), \ldots, \mathcal{R}^{e_k}(f)\}) \tag{7}$$

We observe that under covariate shift, REx is often misaligned with the goal of learning a classifier that only depends on $X_{inv}$. To illustrate this, consider a problem where some instances are more difficult to classify than others (also known as heteroskedastic label noise (Krueger et al., 2021)) and covariate shift results in a case where one training environment has more difficult examples than the other. Then an optimal invariant model would obtain different losses on both environments, meaning it will not satisfy risk invariance. Figure 1 demonstrates this case, and shows that VREx does not learn the invariant classifier.

WRI simplifies to the same problem as VREx when the distribution of invariant features is the same across environments. Yet, while risk invariance only holds in a homoskedastic setting or when there is no invariant feature distribution shift, weighted risk invariance holds in a heteroskedastic setting, regardless of invariant feature distribution shift. Weighted risk invariance can be seen as a reweighting that adjusts samples so environments with different distributions can be compared. Naive reweighting would not account for spurious correlations, but we avoid issues with both covariate shift and spurious correlations by weighting only with the invariant densities.

**Comparison to IRM**   IRM searches for the optimal predictor on latent space that is invariant across environments. This work generated many follow-up extensions (Ahuja et al., 2021; Lu et al., 2021), as well as some critical exploration of its failure cases (Rosenfeld et al., 2020; Ahuja et al., 2020; Kamath et al., 2021; Guo et al., 2021). The original objective is a bi-level optimization problem that is non-convex and challenging to solve, so its relaxation IRMv1 is more often used in practice:

$$\mathcal{R}_{IRM}(f) = \mathcal{R}_{ERM}(f) + \lambda_{IRM}\sum_{e \in \mathcal{E}_{tr}} \|\nabla_{w|w=1.0}\mathcal{R}^e(w \cdot f)\|_2^2 \tag{8}$$

Note that IRM uses the average gradient across the dataset to enforce invariance. This approach inherently introduces a sample complexity issue: when invariance violations are sparse or localized within a large dataset, the average gradient only indirectly accounts for these anomalies, requiring more samples to recognize the violations. Figure 3 shows that for a case of covariate shift in the invariant features, the sample complexity of IRM goes to at least 10000 samples, whereas the WRI method converges to an invariant predictor much earlier. (Note that REx does not converge either— but we do not expect it to converge, regardless of the amount of training data.)

A classical approach to counteract the effects of covariate shift is importance weighting (Shimodaira, 2000). However, a naive application of this method, where the covariates of one environment are

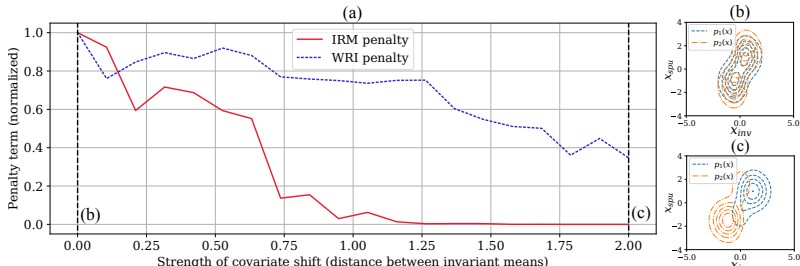

**Figure 4:** (a) We start with two data distributions ($p_1$ and $p_2$) and increase the distance between their means from 0 to 2 in the invariant direction, essentially creating a covariate shift in the invariant features ranging in degree from the diagram in (b) to the diagram in (c). We then search through the set of challenging, slightly spurious classifiers where the ERM loss is close to the loss of the ERM solution, so that we are relying on the invariant penalty to reject these classifiers. We plot how the minimum IRM and WRI penalties in these regions change as covariate shift increases.

reweighted to match the distribution of the other, will not provide the desired result in the scenario of invariant learning.[1]

To better compare WRI and IRM, we revisit the discussion on how the gradient penalty of IRM affects its sample complexity. Suppose there is an area in the feature space where there is no environment overlap at all; that is, suppose there exists a value $c$ where $p_1(x_{inv} = c)$ is large while $p_2(x_{inv} = c) = 0$. A flexible classifier can take advantage of this to use the spurious feature whenever the invariant feature has value $c$ without breaking the invariance constraint (Ahuja et al., 2021). Now, if $p_2(x_{inv} = c) = \varepsilon$ for some small number instead of strictly 0, then a classifier using the spurious feature in that space will violate invariance constraints. In IRM, the violation will correspond to sub-optimality of the learned classifier on top of the representation. However, because sub-optimality is measured using the average gradient over the dataset, the violation will only indirectly contribute to that average, and the sub-optimality will be difficult to detect from a small sample. WRI more directly measures these violations within the average loss, thus it is more sensitive to such violations.

We demonstrate how WRI is more sensitive to such violations in Figure 4. To create this figure, we start with a model that uses spurious features and observe how the IRM and WRI penalties for the model change as we introduce covariate shift to the invariant features. The IRM penalty changes quite significantly as covariate shift increases, while the change for WRI is less pronounced.

## 3.2 IMPLEMENTATION

To evaluate and optimize the weighted risk requires the densities of the invariant feature distributions. However, we do not have access to these—otherwise the invariant learning problem would already be solved. In this section, we propose a method that aims to achieve weighted risk invariance while simultaneously predicting a scaled estimate of the invariant feature densities. We approach this through alternating minimization between a prediction model and environment-specific density models. On one step, we keep the prediction model fixed and update a density model for each environment; on the alternating step, we keep all density models fixed and train the prediction model.

**Practical objective** Minimizing weighted risk invariance recovers a spurious-free predictor, but a WRI term alone is not sufficient to enforce learning true densities (that integrate to 1) over scaled densities (where each pairwise combination is scaled by the same constant), which include the trivial solution where all densities are zero. To regularize the density and prevent the trivial solution, we introduce an additional negative log penalty term that disincentivizes very small density estimates.

---

[1]This occurs because an invariant model on the original distribution may become not-invariant under the reweighted data, where distributions of covariates are matched across environments. It is easy to see this with simple examples.

**Table 1:** Accuracy on HCMNIST without covariate shift and with covariate shift

| Algorithm | HCMNIST | HCMNIST-CS | Avg |
|---|---|---|---|
| ERM | $53.8 \pm 2.3$ | $51.1 \pm 0.9$ | 52.4 |
| IRM | $67.2 \pm 2.7$ | $61.3 \pm 4.7$ | 64.3 |
| VREx | $67.4 \pm 2.3$ | $58.2 \pm 2.3$ | 62.8 |
| WRI | $75.1 \pm 2.9$ | $74.7 \pm 3.8$ | 74.9 |

**Table 2:** Ideal HCMNIST penalty comparison

| Algorithm | Dataset | Penalty ($1e$-3) Digit | Color | Min-penalty classifier |
|---|---|---|---|---|
| WRI | HCMNIST | 0.0 | 4.7 | Digit ✓ |
| | HCMNIST-CS | 0.0 | 0.8 | Digit ✓ |
| VREx | HCMNIST | 0.0 | 5.4 | Digit ✓ |
| | HCMNIST-CS | 1.1 | 0.4 | Color ✗ |

**Table 3:** CMNIST OOD detection

| Algorithm | tpr@fpr=20% | tpr@fpr=40% | tpr@fpr=60% | tpr@fpr=80% | AUROC |
|---|---|---|---|---|---|
| ERM | $25.4 \pm 3.6$ | $45.6 \pm 2.8$ | $64.7 \pm 1.7$ | $82.2 \pm 1.2$ | $0.539 \pm 0.018$ |
| IRM | $29.8 \pm 3.2$ | $46.8 \pm 2.8$ | $66.4 \pm 1.9$ | $84.8 \pm 0.9$ | $0.565 \pm 0.017$ |
| VREx | $25.3 \pm 2.5$ | $43.5 \pm 2.3$ | $61.6 \pm 1.5$ | $80.8 \pm 1.1$ | $0.525 \pm 0.014$ |
| WRI | $36.7 \pm 6.7$ | $54.1 \pm 5.8$ | $68.1 \pm 4.5$ | $85.0 \pm 1.5$ | $0.595 \pm 0.039$ |

To make the dependence of the weighted risk on the invariant feature densities explicit, we write the associated densities $d^{e_j}$ as a parameter of the weighted risk $R^{e_i, e_j}(f; d^{e_j})$. The full loss function is

$$\mathcal{L}(f, \{d^{e_i}\}_{i=1}^k) = \sum_{e \in \mathcal{E}_{tr}} \mathcal{R}^e(f) + \lambda \sum_{\substack{e_i \neq e_j, \\ e_i, e_j \in \mathcal{E}_{tr}}} (\mathcal{R}^{e_i, e_j}(f; d^{e_j}) - \mathcal{R}^{e_j, e_i}(f; d^{e_i}))^2 + \beta \sum_{e \in \mathcal{E}_{tr}} \mathbb{E}_{p_e(X)} \left[ -\log d^e(x) \right]$$

(9)

where $\beta$ is a hyperparameter. Note that this is comprised of the ERM and WRI penalties (as in equation 6), with an additional log penalty term to regularize the learnable density.

Our experiments demonstrate that the densities learned by our implementation are indeed useful for detecting whether an input is part of the training invariant feature distributions or OOD (§4.1). Learning these densities is a key contribution of our method over other domain generalization methods, and the fact that we can learn useful approximations of these validates our alternating minimization approach. For full implementation details, refer to Appendix B.

## 4 EXPERIMENTS

We evaluate our WRI implementation on synthetic and real-world datasets with distribution shifts, particularly focusing on cases of covariate shift in the invariant features. In all of the datasets, we select our model based on a validation set from the test environment, and we report the test set accuracy averaged over 5 random seeds with standard errors. Training validation selection assumes that the training and test data are drawn from similar distributions, which is not the case we aim to be robust to (Gulrajani & Lopez-Paz, 2020); test validation selection is more useful for demonstrating OOD capabilities (Ruan et al., 2021).

Our major baselines are ERM (Vapnik, 1991), IRM (Arjovsky et al., 2019), and VREx (Krueger et al., 2021). IRM and VREx are two other causally-motivated works that also search for conditional invariance as a signature of an underlying causal structure. Because WRI shares a similar theoretical grounding, we find it particularly important to compare our empirical performance with these works. Appendix C includes comparisons with other non-causal baselines, as well as additional experiments and details.

### 4.1 COLOREDMNIST

ColoredMNIST (CMNIST) is a dataset proposed by Arjovsky et al. (2019) as a binary classification extension of MNIST designed to explore the impact of invariant and spurious features on classification. To test the performance of our baselines under the more difficult heteroskedastic setting, we construct a heteroskedastic variant HCMNIST where we vary the label flip probability with the digit. Additionally, we create covariate shift in HCMNIST by creating different distributions of digits in each environment, and we call this dataset HCMNIST-CS (More details on how these datasets were generated can be found in Appendix C.3). Empirical results from these datasets are presented in

**Table 4:** DomainBed results on feature data

| Algorithm | VLCS | PACS | OfficeHome | TerraIncognita | DomainNet | Avg |
|-----------|------|------|------------|----------------|-----------|-----|
| ERM | $76.5 \pm 0.2$ | $84.7 \pm 0.1$ | $64.5 \pm 0.1$ | $51.2 \pm 0.2$ | $33.5 \pm 0.1$ | 62.0 |
| IRM | $76.7 \pm 0.3$ | $84.7 \pm 0.3$ | $63.8 \pm 0.6$ | $52.8 \pm 0.3$ | $22.7 \pm 2.8$ | 60.1 |
| VREx | $76.7 \pm 0.2$ | $84.8 \pm 0.2$ | $64.6 \pm 0.2$ | $52.2 \pm 0.3$ | $26.6 \pm 2.1$ | 61.0 |
| WRI | $77.0 \pm 0.1$ | $85.2 \pm 0.1$ | $64.5 \pm 0.2$ | $52.7 \pm 0.3$ | $32.8 \pm 0.0$ | 62.5 |

Table 1. They demonstrate that WRI consistently performs well in both settings, while VREx and IRM show significant degradation with the introduction of covariate shift.

We also create ideal versions of these datasets where we simplify the image data to two-dimensional features of digit value and color. We evaluate the WRI and VREx penalty terms with a predictor that uses only the digit value (i.e. an invariant predictor) and a predictor that uses only the color (i.e. a spurious predictor), and we report the results in Table 2. As anticipated, WRI registers zero penalty for the invariant predictor in both the HCMNIST and HCMNIST-CS scenarios. In contrast, VREx registers zero penalty for the invariant predictor on HCMNIST but a non-zero penalty on HCMNIST-CS, favoring the spurious predictor under covariate shift. This suggests that the degradation in VREx accuracy under shift in Table 1 can be attributed to a true failure of the VREx penalty to recover the invariant classifier. Additional discussion and details can be found in Appendix C.

**OOD detection performance of our learned densities** To test the estimated invariant density from WRI, we compare its utility as an OOD detector to the model confidences from ERM, IRM, and VREx. The evaluation is conducted on a modified CMNIST test split with mirrored digits. Since the estimated invariant density should be a function of the shape, we expect the invalid digits to have lower density estimates. We compute AUROC values for each experiment and report the results in Table 3. We observe that at all computed false positive rates, the true positive rate is higher for our method; this means that our learned density values are better for detecting OOD digits than the prediction confidences from other methods. For more details, see Appendix C.4

### 4.2 Real-world datasets from DomainBed

We evaluate on 5 real-world datasets that are part of the DomainBed suite, namely VLCS (Fang et al., 2013), PACS (Li et al., 2017), OfficeHome (Venkateswara et al., 2017), TerraIncognita (Beery et al., 2018), and DomainNet (Peng et al., 2019). We run on ERM-trained features for computational efficiency, as these should still contain spurious information as well as the invariant information necessary to generalize OOD (Rosenfeld et al., 2022). In order to achieve a fair comparison, we evaluate all methods starting from the same set of pretrained features. We report the average performance across environments on each dataset in Table 4.

While the WRI predictor achieves higher accuracy than the baselines, we find that the performances are similar. Methods like VREx that rely on certain assumptions may still see high accuracy on datasets that fulfill those assumptions. More information and results are provided in Appendix C.5.

## 5 Related works

**Invariant learning** Outside of causal (and causal-adjacent) literature, much of the theoretical motivation for invariant learning stems from the seminal work of Ben-David et al. (2010), who found that the risk on a test environment is upper bounded by the error on the training environment(s), the total variation between the marginal distributions of training and test, and the difference in labeling functions between training and test. While the first term is minimized in ERM, the second term motivates learning marginally invariant features $\Psi(X)$ such that $p_e(\Psi(X^e))$ is invariant across $e$ (Pan et al., 2010; Baktashmotlagh et al., 2013; Ganin et al., 2016; Tzeng et al., 2017; Long et al., 2018; Zhao et al., 2018), and the third term motivates learning conditionally invariant features $\Phi(X)$ such that $p_e(Y^e|\Phi(X^e))$ is invariant across $e$ (Muandet et al., 2013; Koyama & Yamaguchi, 2020). Observing the importance of both approaches, several works (Zhang et al., 2013; Long et al., 2015; Li et al., 2021) even attempt to learn features that are both marginally and conditionally invariant.

**Density estimation as a measure of uncertainty** Our density estimates serve as an intuitive measure of epistemic uncertainty (Hüllermeier & Waegeman, 2021), as they report how much reliable

(spurious-free) evidence we have in the training data. In general, the uncertainty estimation problem is ill-posed: measures of uncertainty differ between tasks, and even for a specific task there are no ground truth uncertainty estimates (Gawlikowski et al., 2021). Still, providing informative uncertainty estimates is essential to real-world, safety-critical environments such as healthcare (Kompa et al., 2021; Kurz et al., 2022; Mehrtash et al., 2020; Wang et al., 2019) and autonomous driving (Shafaei et al., 2018; Feng et al., 2018; Tang et al., 2022), as well as to the development of other machine learning fields such as sampling in active learning (Yang & Loog, 2016; Shapeev et al., 2020) or balancing exploration/exploitation in reinforcement learning (Kahn et al., 2017; Lütjens et al., 2019). Signatures of good uncertainty estimation include good calibration and OOD detection (Gawlikowski et al., 2021).

**Domain generalization with interpretable signals**  Current invariant learning methods are effective at extracting invariant relationships from training environments, but there are still cases where we cannot utilize these relationships to generalize. Notably for linear classification, Ahuja et al. (2021) proves that generalization is only possible if the support of invariant features in the test environment is a subset of the union of the supports in the training environments. For this reason, we find that learning the density of invariant features from training not only improves our generalization result but also provides a useful signal to tell whether the model is extrapolating from its invariant feature distribution. Quantile Risk Minimization (Eastwood et al., 2022) is another domain generalization method that provides interpretable signals—in their case, with a model selection hyperparameter on the conservativeness of the learned predictor.

## 6 DISCUSSION

**Limitations of our method**  Solving the traditional risk invariance problem does not recover an invariant predictor under heteroskedastic distribution shift, but the addition of an invariant feature density term to WRI allows its solutions to be robust to this case. However, we do not have access to this density in practice, and can only recover an estimate of that density through alternating minimization. Future work could focus on developing algorithms that recover the invariant density with more guarantees, as we believe an accurate invariant density estimate is important for reporting the limitations of domain generalization predictions during deployment.

**Invariant prediction and other approaches**  Invariant prediction is inspired by methods in causal discovery that provide formal guarantees for the identification of causal features (Peters et al., 2016) but are difficult to scale up to complex, high-dimensional data. Methods in invariant prediction focus on making predictions from invariant features rather than learning the invariant features directly. Thus, they do not provide all the guarantees typically offered by methods in causal discovery, but they are faster and simpler in comparison, and can still provide theoretical guarantees for certain causal structures. If causal discovery is first used to extract the invariant features, then works on generalization under covariate shift (Xu et al., 2022; Duchi et al., 2023) can be used to predict from the features under the invariant mechanism assumption.

**Limitations of invariant prediction**  Invariant prediction methods assume there exists some generalizable relationship across training environments, but there are cases where it is impossible to extract the relationship. Notably for linear classification, Ahuja et al. (2021) proves that generalization is only possible if the support of invariant features in the test environment is a subset of the union of the supports in the training environments. For this reason, we think it is important to report when we are given points outside of the invariant feature distribution; then, users know when a domain generalization model is extrapolating outside of training.

## 7 CONCLUSION

We demonstrated the utility of weighted risk invariance for achieving invariant learning, even in the difficult case of heteroskedasticity and covariate shift, and we proved that the solution of WRI recovers a spurious-free predictor under a linear causal setting. We also introduced an algorithm to solve WRI in practice by simultaneously learning the model parameters and the density of the invariant features. In our experiments, we demonstrated that the WRI predictor outperforms state-of-the-art baselines in the case of covariate shift in the invariant features. Finally, we showed that our learned invariant feature density reports when invariant features are outside of the training distribution, letting us know when our predictor is trustworthy.

## 8 REPRODUCIBILITY STATEMENT

We have made significant effort to ensure the figures and results in this paper can be reproduced. The (anonymized) code for generating the figures and empirical results can be found at `https://dl.dropboxusercontent.com/scl/fi/ispvalwz3y196d7b97kke/WRI_code.zip?rlkey=8u5edlx4go1okttmqzsoifiw&dl=0`.

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

## A  PROOFS

**Proposition 1.** *Let Assumption 1 hold over a set of environments $\mathcal{E}'$. If a predictor $f$ is spurious-free over $\mathcal{E}'$, then weighted risk invariance holds over $\mathcal{E}'$.*

*Proof.* Weighted risk between environments 1 and 2 is given by

$$\int_{\mathcal{Y}} \int_{\mathcal{X}_{inv}} \int_{\mathcal{X}_{spu}} p_1(\mathbf{x}_{inv}, \mathbf{x}_{spu}, y) \cdot \ell(f(\mathbf{x}_{inv}, \mathbf{x}_{spu}), y) \cdot p_2(\mathbf{x}_{inv}) \cdot d\,\mathbf{x}_{spu}\ d\,\mathbf{x}_{inv}\ dy.$$

If $f$ is invariant, then we can transform this integral as follows.

$$\int_{\mathcal{Y}} \int_{\mathcal{X}_{inv}} \int_{\mathcal{X}_{spu}} p_1(\mathbf{x}_{inv}, \mathbf{x}_{spu}, y) \cdot \ell(f(\mathbf{x}_{inv}, \mathbf{x}_{spu}), y) \cdot p_2(\mathbf{x}_{inv}) \cdot d\,\mathbf{x}_{spu}\ d\,\mathbf{x}_{inv}\ dy$$

$$= \int_{\mathcal{Y}} \int_{\mathcal{X}_{inv}} \ell(f(\mathbf{x}_{inv}), y) \cdot p_2(\mathbf{x}_{inv}) \int_{\mathcal{X}_{spu}} p_1(\mathbf{x}_{inv}, \mathbf{x}_{spu}, y) \cdot d\,\mathbf{x}_{spu}\ d\,\mathbf{x}_{inv}\ dy$$

$$= \int_{\mathcal{Y}} \int_{\mathcal{X}_{inv}} \ell(f(\mathbf{x}_{inv}), y) \cdot p_2(\mathbf{x}_{inv}) \cdot p_1(\mathbf{x}_{inv}, y)\ d\,\mathbf{x}_{inv}\ dy$$

$$= \int_{\mathcal{Y}} \int_{\mathcal{X}_{inv}} \ell(f(\mathbf{x}_{inv}), y) \cdot p_2(\mathbf{x}_{inv}) \cdot p_1(y|\,\mathbf{x}_{inv}) \cdot p_1(\mathbf{x}_{inv})\ d\,\mathbf{x}_{inv}\ dy. \tag{A.1}$$

Similarly, the weighted risk between environments 2 and 1 can be expressed as

$$\int_{\mathcal{Y}} \int_{\mathcal{X}_{inv}} \ell(f(\mathbf{x}_{inv}), y) \cdot p_1(\mathbf{x}_{inv}) \cdot p_2(y|\,\mathbf{x}_{inv}) \cdot p_2(\mathbf{x}_{inv}) \cdot d\,\mathbf{x}_{inv}\ dy. \tag{A.2}$$

Because $p_e(y|\,\mathbf{x}_{inv})$ is the same for all $e$, then equation A.1 and equation A.2 are equal and weighted risk invariance holds between the two environments. By the same argument, it holds for all pairwise combinations of environments in the set. $\qquad\square$

To show that weighted risk invariance leads to OOD generalization, we must first make an assumption on the diversity of our training environments. We give a definition for general position in a similar style to previous works (Arjovsky et al., 2019; Wald et al., 2021).

**Definition A.1.** We are given a set of $k$ environments such that $\binom{k}{2} > 2d_{spu}$, where $d_{spu}$ is the dimension of spurious features. Each environment $i$ has a *density-weighted* covariance $\Sigma^{i,j}$ and a *density-weighted* correlation $\mu^{i,j}$ with another environment $j$. We define these as

$$\Sigma^{i,j} = \mathbb{E}_{p_i(X^i, Y^i)} \left[ \mathbf{x}\,\mathbf{x}^{\mathsf{T}}\, p_j(\mathbf{x}_{inv}) \right]$$
$$\mu^{i,j} = 2 \cdot \mathbb{E}_{p_i(X^i, Y^i)} \left[ \mathbf{x}\, y p_j(\mathbf{x}_{inv}) \right].$$

We say that this set of environments is in general position if for any scalar $\alpha \in \mathbb{R}$ and all nonzero $\mathbf{x} \in \mathbb{R}^{d_{spu}}$,

$$\dim(\text{span}(\{(\Sigma^{i,j} - \Sigma^{j,i})\,\mathbf{x} + \alpha(\mu^{i,j} - \mu^{j,i})\}_{i,j \in [k]})) = d_{spu}. \tag{A.3}$$

With this definition, we proceed to our main result.

**Theorem 2.** *Consider a regression problem following the data generating process of equation 5. Let $\mathcal{E}'$ be a set of environments that satisfy general position. A linear regression model $f(\mathbf{x}) = \mathbf{w}^{\mathsf{T}}\mathbf{x}$ with $\mathbf{w} = [\mathbf{w}_{inv}, \mathbf{w}_{spu}]$ that satisfies weighted risk invariance w.r.t the squared loss must satisfy $\mathbf{w}_{spu} = 0$.*

*Proof.* Assume we have weighted risk invariance between environments 1 and 2, so

$$\int_{\mathcal{Y}} \int_{\mathcal{X}_{inv}} \int_{\mathcal{X}_{spu}} (\mathbf{w}^{\mathsf{T}}[\mathbf{x}_{inv}, \mathbf{x}_{spu}] - y)^2 \cdot p_1(\mathbf{x}_{inv}, \mathbf{x}_{spu}, y) \cdot p_2(\mathbf{x}_{inv}) \cdot d\,\mathbf{x}_{spu}\ d\,\mathbf{x}_{inv}\ dy$$

$$= \int_{\mathcal{Y}} \int_{\mathcal{X}_{inv}} \int_{\mathcal{X}_{spu}} (\mathbf{w}^{\mathsf{T}}[\mathbf{x}_{inv}, \mathbf{x}_{spu}] - y)^2 \cdot p_2(\mathbf{x}_{inv}, \mathbf{x}_{spu}, y) \cdot p_1(\mathbf{x}_{inv}) \cdot d\,\mathbf{x}_{spu}\ d\,\mathbf{x}_{inv}\ dy. \tag{A.4}$$

When we expand the square, the left hand side becomes

$$
\mathbf{w}^{\mathsf{T}} \left( \int_{\mathcal{Y}} \int_{\mathcal{X}_{inv}} \int_{\mathcal{X}_{spu}} [\mathbf{x}_{inv}, \mathbf{x}_{spu}][\mathbf{x}_{inv}, \mathbf{x}_{spu}]^{\mathsf{T}} \cdot p_1(\mathbf{x}_{inv}, \mathbf{x}_{spu}, y) \cdot p_2(\mathbf{x}_{inv}) \cdot d\,\mathbf{x}_{spu} \; d\,\mathbf{x}_{inv} \; dy \right) \mathbf{w}
$$
$$
- 2\,\mathbf{w}^{\mathsf{T}} \int_{\mathcal{Y}} \int_{\mathcal{X}_{inv}} \int_{\mathcal{X}_{spu}} [\mathbf{x}_{inv}, \mathbf{x}_{spu}]y \cdot p_1(\mathbf{x}_{inv}, \mathbf{x}_{spu}, y) \cdot p_2(\mathbf{x}_{inv}) \cdot d\,\mathbf{x}_{spu} \; d\,\mathbf{x}_{inv} \; dy
$$
$$
+ \int_{\mathcal{Y}} \int_{\mathcal{X}_{inv}} \int_{\mathcal{X}_{spu}} y^2 \cdot p_1(\mathbf{x}_{inv}, \mathbf{x}_{spu}, y) \cdot p_2(\mathbf{x}_{inv}) \cdot d\,\mathbf{x}_{spu} \; d\,\mathbf{x}_{inv} \; dy.
$$

We define

$$
\Sigma^{1,2} = \int_{\mathcal{Y}} \int_{\mathcal{X}_{inv}} \int_{\mathcal{X}_{spu}} [\mathbf{x}_{inv}, \mathbf{x}_{spu}][\mathbf{x}_{inv}, \mathbf{x}_{spu}]^{\mathsf{T}} \cdot p_1(\mathbf{x}_{inv}, \mathbf{x}_{spu}, y) \cdot p_2(\mathbf{x}_{inv}) \cdot d\,\mathbf{x}_{spu} \; d\,\mathbf{x}_{inv} \; dy
$$
$$
= \int_{\mathcal{X}_{inv}} \int_{\mathcal{X}_{spu}} [\mathbf{x}_{inv}, \mathbf{x}_{spu}][\mathbf{x}_{inv}, \mathbf{x}_{spu}]^{\mathsf{T}} \cdot p_1(\mathbf{x}_{inv}, \mathbf{x}_{spu}) \cdot p_2(\mathbf{x}_{inv}) \cdot d\,\mathbf{x}_{spu} \; d\,\mathbf{x}_{inv}
$$
$$
\mu^{1,2} = 2 \int_{\mathcal{Y}} \int_{\mathcal{X}_{inv}} \int_{\mathcal{X}_{spu}} [\mathbf{x}_{inv}, \mathbf{x}_{spu}]y \cdot p_1(\mathbf{x}_{inv}, \mathbf{x}_{spu}, y) \cdot p_2(\mathbf{x}_{inv}) \cdot d\,\mathbf{x}_{spu} \; d\,\mathbf{x}_{inv} \; dy
$$
$$
c^{1,2} = \int_{\mathcal{Y}} \int_{\mathcal{X}_{inv}} \int_{\mathcal{X}_{spu}} y^2 \cdot p_1(\mathbf{x}_{inv}, \mathbf{x}_{spu}, y) \cdot p_2(\mathbf{x}_{inv}) \cdot d\,\mathbf{x}_{spu} \; d\,\mathbf{x}_{inv} \; dy
$$
$$
= \int_{\mathcal{Y}} \int_{\mathcal{X}_{inv}} y^2 \cdot p_1(\mathbf{x}_{inv}, y) \cdot p_2(\mathbf{x}_{inv}) \cdot d\,\mathbf{x}_{inv} \; dy,
$$

with analogous definitions for $\Sigma^{2,1}$, $\mu^{2,1}$, and $c^{2,1}$. Then equation A.4 can be written more succinctly as

$$
\mathbf{w}^{\mathsf{T}} \Sigma^{1,2} \, \mathbf{w} - \mathbf{w}^{\mathsf{T}} \mu^{1,2} + c^{1,2} = \mathbf{w}^{\mathsf{T}} \Sigma^{2,1} \, \mathbf{w} - \mathbf{w}^{\mathsf{T}} \mu^{2,1} + c^{2,1}. \tag{A.5}
$$

We decompose the quadratic term $\Sigma^{1,2}$ into invariant, spurious, and mixed components:

$$
\Sigma^{1,2} = \int_{\mathcal{X}_{inv}} \int_{\mathcal{X}_{spu}} [\mathbf{x}_{inv}, \mathbf{x}_{spu}][\mathbf{x}_{inv}, \mathbf{x}_{spu}]^{\mathsf{T}} \cdot p_1(\mathbf{x}_{inv}, \mathbf{x}_{spu}) \cdot p_2(\mathbf{x}_{inv}) \cdot d\,\mathbf{x}_{spu} \; d\,\mathbf{x}_{inv}
$$
$$
= \int_{\mathcal{X}_{inv}} \int_{\mathcal{X}_{spu}} \begin{bmatrix} \mathbf{x}_{inv}\,\mathbf{x}_{inv}^{\mathsf{T}} & \mathbf{x}_{inv}\,\mathbf{x}_{spu}^{\mathsf{T}} \\ \mathbf{x}_{spu}\,\mathbf{x}_{inv}^{\mathsf{T}} & \mathbf{x}_{spu}\,\mathbf{x}_{spu}^{\mathsf{T}} \end{bmatrix} \cdot p_1(\mathbf{x}_{inv}, \mathbf{x}_{spu}) \cdot p_2(\mathbf{x}_{inv}) \cdot d\,\mathbf{x}_{spu} \; d\,\mathbf{x}_{inv}
$$
$$
= \int_{\mathcal{X}_{inv}} \begin{bmatrix} \mathbf{x}_{inv}\,\mathbf{x}_{inv}^{\mathsf{T}} & \mathbf{0} \\ \mathbf{0} & \mathbf{0} \end{bmatrix} \cdot p_1(\mathbf{x}_{inv}) \cdot p_2(\mathbf{x}_{inv}) \cdot d\,\mathbf{x}_{inv}
$$
$$
+ \int_{\mathcal{X}_{inv}} \int_{\mathcal{X}_{spu}} \begin{bmatrix} \mathbf{0} & \mathbf{0} \\ \mathbf{0} & \mathbf{x}_{spu}\,\mathbf{x}_{spu}^{\mathsf{T}} \end{bmatrix} \cdot p_1(\mathbf{x}_{inv}, \mathbf{x}_{spu}) \cdot p_2(\mathbf{x}_{inv}) \cdot d\,\mathbf{x}_{spu} \; d\,\mathbf{x}_{inv}.
$$
$$
+ \int_{\mathcal{X}_{inv}} \int_{\mathcal{X}_{spu}} \begin{bmatrix} \mathbf{0} & \mathbf{x}_{inv}\,\mathbf{x}_{spu}^{\mathsf{T}} \\ \mathbf{x}_{spu}\,\mathbf{x}_{inv}^{\mathsf{T}} & \mathbf{0} \end{bmatrix} \cdot p_1(\mathbf{x}_{inv}, \mathbf{x}_{spu}) \cdot p_2(\mathbf{x}_{inv}) \cdot d\,\mathbf{x}_{spu} \; d\,\mathbf{x}_{inv}.
$$
$$
= \begin{bmatrix} \Sigma_{inv}^{1,2} & \mathbf{0} \\ \mathbf{0} & \mathbf{0} \end{bmatrix} + \begin{bmatrix} \mathbf{0} & \mathbf{0} \\ \mathbf{0} & \Sigma_{spu}^{1,2} \end{bmatrix} + \Sigma_{mix}^{1,2}.
$$

We break down the third term $\Sigma_{mix}^{1,2}$ in more detail, following the data-generating process for $\mathbf{x}_{spu}$.

$$
\begin{aligned}
\Sigma_{mix}^{1,2} &= \int_{\mathcal{X}_{inv}} \int_{\mathcal{X}_{spu}} \begin{bmatrix} \mathbf{0} & \mathbf{x}_{inv}\,\mathbf{x}_{spu}^{\mathsf{T}} \\ \mathbf{x}_{spu}\,\mathbf{x}_{inv}^{\mathsf{T}} & \mathbf{0} \end{bmatrix} \cdot p_1(\mathbf{x}_{inv}, \mathbf{x}_{spu}) \cdot p_2(\mathbf{x}_{inv}) \cdot d\,\mathbf{x}_{spu}\ d\,\mathbf{x}_{inv} \\
&= \int_{\mathcal{X}_{inv}} \int_{\varepsilon} \int_{\eta} \begin{bmatrix} \mathbf{0} & \mathbf{x}_{inv}\,\mu_i^{\mathsf{T}}\,\mathbf{x}_{inv}^{\mathsf{T}}\,\mathbf{w}_{inv}^{*} + \mathbf{x}_{inv}\,\mu_i^{\mathsf{T}}\varepsilon^{\mathsf{T}} + \mathbf{x}_{inv}\,\eta^{\mathsf{T}} \\ (\mathbf{w}_{inv}^{*})^{\mathsf{T}}\,\mathbf{x}_{inv}\,\mu_i\,\mathbf{x}_{inv}^{\mathsf{T}} + \varepsilon\mu_i\,\mathbf{x}_{inv}^{\mathsf{T}} + \eta\,\mathbf{x}_{inv}^{\mathsf{T}} & \mathbf{0} \end{bmatrix} \cdots \\
&\qquad \cdots p_1(\varepsilon) \cdot p_1(\eta) \cdot p_1(\mathbf{x}_{inv}) \cdot p_2(\mathbf{x}_{inv}) \cdot d\eta\ d\varepsilon\ d\,\mathbf{x}_{inv} \\
&= \int_{\mathcal{X}_{inv}} \begin{bmatrix} \mathbf{0} & \mathbf{x}_{inv}\,\mu_i^{\mathsf{T}}\,\mathbf{x}_{inv}^{\mathsf{T}}\,\mathbf{w}_{inv}^{*} \\ (\mathbf{w}_{inv}^{*})^{\mathsf{T}}\,\mathbf{x}_{inv}\,\mu_i\,\mathbf{x}_{inv}^{\mathsf{T}} & \mathbf{0} \end{bmatrix} \cdot p_1(\mathbf{x}_{inv}) \cdot p_2(\mathbf{x}_{inv}) \cdot d\,\mathbf{x}_{inv} \\
&\quad + \int_{\mathcal{X}_{inv}} \int_{\varepsilon} \begin{bmatrix} \mathbf{0} & \mathbf{x}_{inv}\,\mu_i^{\mathsf{T}} \\ \mu_i\,\mathbf{x}_{inv}^{\mathsf{T}} & \mathbf{0} \end{bmatrix} \cdot \varepsilon \cdot p_1(\varepsilon) \cdot p_1(\mathbf{x}_{inv}) \cdot p_2(\mathbf{x}_{inv}) \cdot d\varepsilon\ d\,\mathbf{x}_{inv} \\
&\quad + \int_{\mathcal{X}_{inv}} \int_{\eta} \begin{bmatrix} \mathbf{0} & \mathbf{x}_{inv}\,\eta^{\mathsf{T}} \\ \eta\,\mathbf{x}_{inv}^{\mathsf{T}} & \mathbf{0} \end{bmatrix} \cdot p_1(\eta) \cdot p_1(\mathbf{x}_{inv}) \cdot p_2(\mathbf{x}_{inv}) \cdot d\eta\ d\,\mathbf{x}_{inv}\,.
\end{aligned}
$$

Both $\varepsilon$ and $\eta$ are mean-zero random variables that are independent of $\mathbf{x}_{inv}$. This means the second and third terms go to zero, leaving us with only the first term. We further decompose the first term, now explicitly labeling the dimensions of the 0 submatrices for clarity.

$$
\Sigma_{mix}^{1,2} = \begin{bmatrix} I_{d_{inv}} & \mathbf{0}_{d_{inv}\times 1} \\ \mathbf{0}_{d_{spu}\times d_{inv}} & \mu_i \end{bmatrix} \Sigma_{w-inv}^{1,2} \begin{bmatrix} I_{d_{inv}} & \mathbf{0}_{d_{inv}\times d_{spu}} \\ \mathbf{0}_{1\times d_{inv}} & \mu_i^{\mathsf{T}} \end{bmatrix},
$$

where

$$
\begin{aligned}
\Sigma_{w-inv}^{1,2} &= \int_{\mathcal{X}_{inv}} \begin{bmatrix} \mathbf{0}_{d_{inv}\times d_{inv}} & \mathbf{x}_{inv}(\mathbf{w}_{inv}^{*})^{\mathsf{T}}\,\mathbf{x}_{inv} \\ \mathbf{x}_{inv}^{\mathsf{T}}\,\mathbf{w}_{inv}^{*}\,\mathbf{x}_{inv}^{\mathsf{T}} & \mathbf{0}_{1\times 1} \end{bmatrix} \cdot p_1(\mathbf{x}_{inv}) \cdot p_2(\mathbf{x}_{inv}) \cdot d\,\mathbf{x}_{inv} \\
&= \begin{bmatrix} \mathbf{0}_{d_{inv}\times d_{inv}} & \mathbf{a}^{1,2} \\ (\mathbf{a}^{1,2})^{\mathsf{T}} & \mathbf{0}_{1\times 1} \end{bmatrix},
\end{aligned}
$$

with $\mathbf{a}^{1,2} = \mathbf{x}_{inv}(\mathbf{w}_{inv}^{*})^{\mathsf{T}}\,\mathbf{x}_{inv}$ being a $d_{inv}$-dimensional column vector.

We also separate the linear term $\mu^{1,2}$ into invariant and spurious components:

$$
\begin{aligned}
\mu^{1,2} =\ &2 \int_{\mathcal{Y}} \int_{\mathcal{X}_{inv}} \int_{\mathcal{X}_{spu}} [\mathbf{x}_{inv}, \mathbf{x}_{spu}]y \cdot p_1(\mathbf{x}_{inv}, \mathbf{x}_{spu}, y) \cdot p_2(\mathbf{x}_{inv}) \cdot d\,\mathbf{x}_{spu}\ d\,\mathbf{x}_{inv}\ dy \\
=\ &2 \int_{\mathcal{Y}} \int_{\mathcal{X}_{inv}} [\mathbf{x}_{inv}, \mathbf{0}]y \int_{\mathcal{X}_{spu}} p_1(\mathbf{x}_{inv}, \mathbf{x}_{spu}, y) \cdot p_2(\mathbf{x}_{inv}) \cdot d\,\mathbf{x}_{spu}\ d\,\mathbf{x}_{inv}\ dy \\
&+ 2 \int_{\mathcal{Y}} \int_{\mathcal{X}_{spu}} [\mathbf{0}, \mathbf{x}_{spu}]y \int_{\mathcal{X}_{inv}} p_1(\mathbf{x}_{inv}, \mathbf{x}_{spu}, y) \cdot p_2(\mathbf{x}_{inv}) \cdot d\,\mathbf{x}_{inv}\ d\,\mathbf{x}_{spu}\ dy \\
=\ &2 \int_{\mathcal{Y}} \int_{\mathcal{X}_{inv}} [\mathbf{x}_{inv}, \mathbf{0}]y \cdot p_1(y|\,\mathbf{x}_{inv}) \cdot p_1(\mathbf{x}_{inv}) \cdot p_2(\mathbf{x}_{inv}) \cdot d\,\mathbf{x}_{inv}\ dy \\
&+ 2 \int_{\mathcal{Y}} \int_{\mathcal{X}_{spu}} [\mathbf{0}, \mathbf{x}_{spu}]y \int_{\mathcal{X}_{inv}} p_1(\mathbf{x}_{inv}, \mathbf{x}_{spu}, y) \cdot p_2(\mathbf{x}_{inv}) \cdot d\,\mathbf{x}_{inv}\ d\,\mathbf{x}_{spu}\ dy \\
=\ &[\mu_{inv}^{1,2}, \mathbf{0}] + [\mathbf{0}, \mu_{spu}^{1,2}].
\end{aligned}
$$

Finally, we break down the constant term $c^{1,2}$.

$$
\begin{aligned}
c^{1,2} &= \int_{\mathcal{Y}} \int_{\mathcal{X}_{inv}} y^2 \cdot p_1(\mathbf{x}_{inv}, y) \cdot p_2(\mathbf{x}_{inv}) \cdot d\,\mathbf{x}_{inv}\ dy \\
&= \int_{\mathcal{Y}} \int_{\mathcal{X}} y^2 \cdot p_1(y|\,\mathbf{x}_{inv}) \cdot p_1(\mathbf{x}_{inv}) \cdot p_2(\mathbf{x}_{inv}) \cdot d\,\mathbf{x}_{inv}\ dy.
\end{aligned}
$$

It is clear that $\Sigma_{inv}^{1,2} = \Sigma_{inv}^{2,1}$, $\mathbf{a}^{1,2} = \mathbf{a}^{2,1}$, $\mu_{inv}^{1,2} = \mu_{inv}^{2,1}$, and $c^{1,2} = c^{2,1}$. With this information, we define $\alpha := 2\,\mathbf{w}_{inv}^{\mathsf{T}}\,\mathbf{a}^{1,2} - 1$ and simplify equation A.5 to

$$\mathbf{w}_{spu}^{\mathsf{T}} \Sigma_{spu}^{1,2} \mathbf{w}_{spu} + \alpha(\mu_{spu}^{1,2})^{\mathsf{T}} \mathbf{w}_{spu} = \mathbf{w}_{spu}^{\mathsf{T}} \Sigma_{spu}^{2,1} \mathbf{w}_{spu} + \alpha(\mu_{spu}^{2,1})^{\mathsf{T}} \mathbf{w}_{spu}. \tag{A.6}$$

Weighted risk invariance equation A.4 holds across a set of environments if and only if equation A.6 holds for each pairwise combination of environments. Further, if equation A.6 holds for each pairwise combination of environments, then

$$\mathbf{w}_{spu}^{\mathsf{T}}(\Sigma_{spu}^{i,j} - \Sigma_{spu}^{j,i}) \mathbf{w}_{spu} + \alpha(\mu_{spu}^{i,j} - \mu_{spu}^{j,i})^{\mathsf{T}} \mathbf{w}_{spu} = 0 \quad \forall i,j \in [k]. \tag{A.7}$$

We define the $\binom{k}{2} \times d_{spu}$ matrix

$$M = \begin{bmatrix} \mathbf{w}_{spu}^{\mathsf{T}}(\Sigma_{spu}^{1,2} - \Sigma_{spu}^{2,1}) + \alpha(\mu_{spu}^{1,2} - \mu_{spu}^{2,1})^{\mathsf{T}} \\ \mathbf{w}_{spu}^{\mathsf{T}}(\Sigma_{spu}^{1,3} - \Sigma_{spu}^{3,1}) + \alpha(\mu_{spu}^{1,3} - \mu_{spu}^{3,1})^{\mathsf{T}} \\ \vdots \\ \mathbf{w}_{spu}^{\mathsf{T}}(\Sigma_{spu}^{k,k-1} - \Sigma_{spu}^{k-1,k}) + \alpha(\mu_{spu}^{k,k-1} - \mu_{spu}^{k-1,k})^{\mathsf{T}} \end{bmatrix}.$$

Since the environments are in general position, then matrix $M$ is full rank for any nonzero $\mathbf{w}_{spu}$. That means that there is no nonzero vector $\mathbf{x}$ that solves

$$M\mathbf{x} = \mathbf{0}. \tag{A.8}$$

If there is no nonzero solution to equation A.8, then there is no nonzero $\mathbf{w}_{spu}$ that solves equation A.7. Thus, weighted risk invariance equation A.5 implies that $\mathbf{w}_{spu} = 0$. □

---

**Algorithm B.1:** WRI with model-based density

---

**Parameters:**
    $n$: total number of optimization steps
    $\omega$: density update frequency
    $n_d$: number of density update steps

**initialize** random model weights for $f$ and $d^e$ for all $e \in \mathcal{E}_{tr}$
**initialize** Adam (Kingma & Ba, 2017) optimizer for $f$
**initialize** Adam optimizer for $d^e$
**for** $i = 1 \ldots n$ **do**
    sample featurized minibatch from all environments
    compute $\mathcal{L}$ equation 9 on minibatch and step $f$ optimizer
    **if** $i - 1$ is a multiple of $\omega$ **then**
        **repeat** $n_d$ **times**
            sample featurized minibatch from all environments
            compute $\mathcal{L}$ on minibatch and step $d^e$ optimizer

---

## B   Implementation details

When the optimal invariant predictor was first introduced with IRM, it was first motivated as a constrained optimization problem that searched over all spurious-free predictors for the predictor with the lowest training loss. The practical objective IRMv1 was then introduced, where the constraint was approximated by a gradient penalty. Many follow-up works took a similar path, where a penalty is used as some form of invariance constraint. Our penalty is weighted risk invariance; according to our theory, models with zero weights on spurious features will satisfy the WRI constraint. Therefore, a learning rule that finds the most accurate classifier satisfying WRI will learn an optimal hypothesis that relies on the invariant features alone.

We implement this learning rule with alternating minimization. This is a complex optimization procedure, and we do not derive guarantees on its convergence to invariant representations in this work. We simply motivate this implementation with the knowledge that a spurious-free predictor would minimize the WRI penalty (which we optimize for when the penalty coefficient in equation 9 is sufficiently large). We test that this objective is effective at recovering a generalizable predictor (as well as meaningful OOD invariance density estimates) empirically. Our specific algorithm is described in Algorithm B.1.

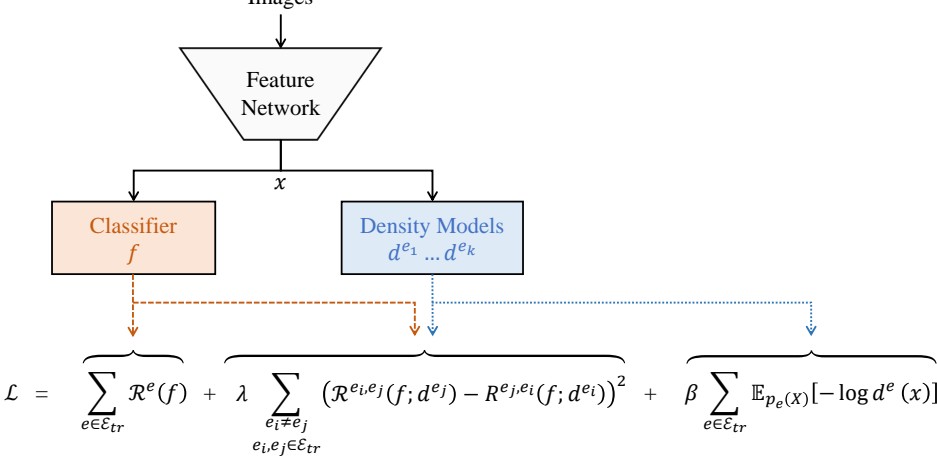

**Figure B.1:** Graphic overview of our network architecture. The red dashed line indicates the components that impact $f$ when taking an optimization step. The blue dotted line indicates the components that affect the density models when they are updated.

We observe that the magnitude of the WRI regularization term from equation 9 varies significantly in practice, making it difficult to choose a good value for $\lambda$. This is true even when the true invariant density is known. We employ two strategies to deal with this. First, we constrain the density model to predict values within a pre-defined range by applying a sigmoid activation on the final prediction with constant scale and shift factor. Additionally, we divide the WRI term by the average negative log-likelihood, which also helps to decouple the empirical risk from the WRI regularization term.

We allow different optimization parameters for the prediction model and density estimation models. Both optimizers have a different learning rate, weight decay, batch size, and $\lambda$ penalty (for the WRI penalty). The range of hyperparameter values we use are provided with the DomainBed details in Appendix C.5.

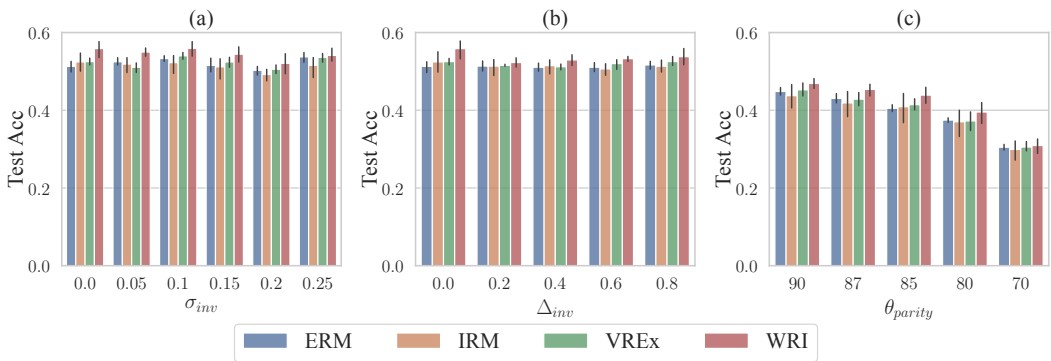

**Figure C.2:** Our simulated classification setting from §C.2 where we measure the test accuracy under different shifts in the invariant features. (a) We vary the average distance between invariant distributions. (b) We vary the covariance matrices between invariant distributions. (c) With $\sigma_{inv} = 0.15$, $\Delta_{inv} = 0.5$, we vary the correlation of spurious features between the training environments with test having the opposite correlation. In all cases, the WRI predictor outperforms other methods on average.

## C  EXTENSIONS ON EXPERIMENTS AND ADDITIONAL EXPERIMENTAL DETAILS

### C.1  TOY DATASET HYPERPARAMETERS (FIGURE 1 DETAILS)

To produce Figure 1, we define a toy dataset consisting of the three environments described here. Environments 1 and 2 are used for training and environment 3 is used as test. We let $Y = \mathbb{1}[X_{inv} + \varepsilon > 0]$, where $\varepsilon$ is a standard normal random variable and $\mathbb{1}$ is the indicator function. The following data distributions are used:

$$X_{inv}^1 \sim N(0, 2^2), \qquad X_{inv}^2 \sim N(0, \left(\tfrac{1}{2}\right)^2), \qquad X_{inv}^3 \sim N(0, 3^2),$$
$$X_{spu}^1 | Y = 0 \sim N(1, \left(\tfrac{1}{2}\right)^2), \quad X_{spu}^2 | Y = 0 \sim N(1, 2^2), \qquad X_{spu}^3 | Y = 0 \sim N(-1, 1^2),$$
$$X_{spu}^1 | Y = 1 \sim N(-1, \left(\tfrac{1}{2}\right)^2), \quad X_{spu}^2 | Y = 1 \sim N(-1, 2^2), \text{ and} \quad X_{spu}^3 | Y = 1 \sim N(1, 1^2).$$

We sample $10^4$ points from each environment. The predictor is a simple linear model. The ERM, IRM, VREx, and WRI objectives are optimized using scikit-learn (Pedregosa et al., 2011). We use a penalty weight of $10^5$ for IRM, a penalty weight of 1 for VREx, and a penalty weight of 1500 for WRI. We note that VREx converges to an absurd solution when using a larger weight, despite generally using a much larger weight in the DomainBed implementation. The penalty weights were roughly optimized by eye to achieve the best qualitative results (regarding invariance).

### C.2  EXPERIMENTS ON MULTI-DIMENSIONAL SYNTHETIC DATASET

**Data generated following structural equation model**   We construct a multi-class classification simulation based on the linear causal model shown in Figure 2. The spurious and invariant distributions are drawn from normal distributions $X_{inv} \sim N(\mu^e, \Sigma^e)$ and $X_{spu}^e | Y = y \sim N(\mu^{e,y}, \Sigma^{e,y})$ and $X$ is the concatenation of $X_{inv}$ and $X_{spu}$. To simulate the classification scenario, we define the class label to be $Y = \arg\max_y \mathbf{w}_y^\mathsf{T} \mathbf{x}_{inv} + \varepsilon_y$ where $\varepsilon_y \sim N(0, \sigma_y)$.

Rather than specifying all of the distribution parameters manually, we sample them randomly according to

$$\mu^e \sim N(0, I\sigma_{inv}^2), \qquad \Sigma^e = RandCov(1 - \Delta_{inv}, 1 + \Delta_{inv}), \qquad \mathbf{w}_y \sim U(\mathbb{S}_{d_{inv}}),$$
$$\mu^{e,y} \sim N(0, I\sigma_{spu}^2), \qquad \Sigma^{e,y} = RandCov(\tfrac{1}{2}, \tfrac{3}{2}), \qquad \sigma_y = \sigma_y \tag{C.9}$$

where $\mathbb{S}_{d_{inv}}$ is the $d_{inv}$-dimensional hypersphere, and $RandCov(a, b)$ is a covariance matrix-generating random variable that selects the square root of the eigenvalues i.i.d. from $U(a, b)$.

This formulation allows us to specify the data simulation using four scalar values: $\sigma_{inv}, \sigma_{spu}, \Delta_{inv}$, and $\sigma_y$. The $\sigma_{inv}$ and $\sigma_{spu}$ terms define the variability of the means between different invariant and

spurious distributions respectively. For example, when $\sigma_{inv}$ is zero, $X_{inv}$ will have the same mean value across all environments. $\Delta_{inv}$ is a value between 0 and 1 that defines how much the invariant covariance matrices vary between environments, and $\sigma_y$ specifies the noise in the label generation process.

Figure C.2 shows how we compare to other baselines when we run on data with 20 dimensions ($d_{inv} = 10$), with 4 training environments and 1 test. We see that (a) with isotropic covariance, we outperform other baselines when the invariant distribution means overlap and when we increase the distance between the means. Further, (b) with identical means, we outperform other baselines when the invariant distributions all have isotropic covariance and when the covariance matrices vary between distributions. Both cases represent the covariate shift in the invariant features, which our predictor is designed to be more robust to. On average, our improvement over the baselines is more significant when we have isotropic covariance for all environments and increase the distance between the environment means; we believe this is because the isotropic covariance allows for a clearer (more controlled) case of covariate shift.

**Testing dependence on controlled spurious correlation** We run a modified version of our simulation that tests the misleading nature of the spurious features. Similar to the ColoredMNIST approach of defining the train and test environments to have opposite color-label correlation, we randomly select a vector for each class and ensure that its correlation with the mean of the spurious distributions is positive for each training environment and negative for the test environment. This is accomplished by first sampling the data as in equation C.9, flipping correlation if necessary, then applying a transform to ensure that the means are in a hypercone with opening angle $\theta_{parity}$. As $\theta_{parity}$ decreases from $90°$, the spurious data increasingly predicts $Y$ consistently across training environments, making it more challenging to disentangle the spurious and invariant features. In Figure C.2(c), we see that although decreasing $\theta_{parity}$ degrades the performance of all methods, WRI still performs better than the other baselines.

**Additional discussion of simulation hyperparameters** Here, we further discuss the hyperparameters $\sigma_{inv}$, $\Delta_{inv}$, $\sigma_y$, and $\sigma_{spu}$, and how they impact the data generation.

$\sigma_{inv}$ is a non-negative scalar that encapsulates the distance between invariant data for each environment. When this value is zero, the invariant distributions are all centered on the origin; when this value is large enough, the average distance between invariant distributions increases.

$\Delta_{inv}$ is a scalar in $[0, 1]$ that encapsulates how much the covariance matrices of the invariant distributions differ between environments. When this value is zero, all environments have identity covariance. Intuitively, this value is similar to $\sigma_{inv}$ in the sense that larger values will produce invariant distributions that overlap less between environments.

$\sigma_y$ is the only directly specified parameter of the data model and it controls the amount of label noise present. When set to zero, the label is a deterministic function of $X_{inv}$; when set very high, $Y$ becomes more difficult to predict from the invariant data.

$\sigma_{spu}$ is a non-negative scalar that encapsulates the distance between spurious data distributions. As this value gets larger, the spurious distributions are less likely to overlap both between classes and between environments. This has two primary effects. First, within a single environment, less overlap will make spurious data more predictive of the label. However, because the locations of the distributions change between environments, this also means that the spurious nature of $X_{spu}$ should be more apparent during training.

One caveat to our method of data generation is that controlling for a uniform number of classes is difficult since the class label is a function of $X_{inv}$. For this reason, we sample a new set of data model parameters when one class is overrepresented in any environment by a factor of 1.5 times the expected uniform representation (e.g. for 5 classes, we resample when more than 30% of data from any environment have the same label). This also ensures that any models with accuracy above 1.5 times uniform are making non-trivial predictions.

**Controlling the spurious correlation** For the experiment reported in Figure C.2 (c), we explicitly manipulate the data model to ensure a higher level of spurious correlation via the $\theta_{parity}$ hyperparameter. This produces a scenario similar to ColoredMNIST, where the correlation between the label and spurious data (color) is relatively consistent between training environments ($+90\%$ and $+80\%$) but has the opposite relationship in test ($-90\%$). This means that an algorithm that learns

**Table C.1:** Sweep $\sigma_{inv}$ ($\Delta_{inv} = 0, \sigma_y = 0.5, \sigma_{spu} = 1$). Plotted in Figure C.2 (a)

| Algorithm | $\sigma_{inv}$ | | | | | | |
| | 0.0 | 0.05 | 0.1 | 0.15 | 0.2 | 0.25 | 0.3 |
|---|---|---|---|---|---|---|---|
| ERM | $51.3 \pm 1.6$ | $52.5 \pm 1.1$ | $53.3 \pm 0.9$ | $51.5 \pm 2.0$ | $50.3 \pm 1.3$ | $53.7 \pm 1.4$ | $54.8 \pm 2.1$ |
| IRM | $52.4 \pm 2.7$ | $51.9 \pm 2.3$ | $52.2 \pm 2.7$ | $51.2 \pm 3.2$ | $49.2 \pm 1.8$ | $51.5 \pm 3.2$ | $54.3 \pm 3.3$ |
| VREx | $52.5 \pm 1.0$ | $51.1 \pm 1.5$ | $54.0 \pm 1.0$ | $52.4 \pm 1.4$ | $50.5 \pm 1.3$ | $53.6 \pm 1.3$ | $56.8 \pm 1.1$ |
| WRI | $55.9 \pm 2.5$ | $55.0 \pm 1.3$ | $55.9 \pm 2.3$ | $54.4 \pm 2.3$ | $52.1 \pm 3.1$ | $54.1 \pm 1.8$ | $56.1 \pm 2.6$ |

**Table C.2:** Sweep $\Delta_{inv}$ ($\sigma_{inv} = 0, \sigma_y = 0.5, \sigma_{spu} = 1$). Plotted in Figure C.2 (b)

| Algorithm | $\Delta_{inv}$ | | | | |
| | 0.0 | 0.2 | 0.4 | 0.6 | 0.8 |
|---|---|---|---|---|---|
| ERM | $51.3 \pm 1.6$ | $51.4 \pm 1.5$ | $51.1 \pm 1.2$ | $51.1 \pm 1.3$ | $51.7 \pm 1.2$ |
| IRM | $52.4 \pm 2.7$ | $51.3 \pm 2.4$ | $51.5 \pm 2.1$ | $50.6 \pm 1.7$ | $51.3 \pm 1.8$ |
| VREx | $52.5 \pm 1.0$ | $51.6 \pm 0.3$ | $51.2 \pm 0.9$ | $52.0 \pm 1.5$ | $52.5 \pm 1.3$ |
| WRI | $55.9 \pm 2.5$ | $52.3 \pm 1.4$ | $52.9 \pm 1.6$ | $53.3 \pm 0.7$ | $53.8 \pm 2.5$ |

**Table C.3:** Sweep $\theta_{parity}$ ($\sigma_{inv} = 0.15, \Delta_{inv} = 0.5, \sigma_y = 0.5, \sigma_{spu} = 1$). Plotted in Figure C.2 (c)

| Algorithm | $\theta_{parity}$ | | | | |
| | 90° | 87° | 85° | 80° | 70° |
|---|---|---|---|---|---|
| ERM | $44.8 \pm 1.2$ | $43.0 \pm 1.4$ | $40.5 \pm 1.1$ | $37.5 \pm 0.7$ | $30.5 \pm 0.9$ |
| IRM | $43.8 \pm 3.6$ | $41.9 \pm 3.9$ | $40.9 \pm 4.2$ | $37.0 \pm 4.0$ | $29.9 \pm 3.0$ |
| VREx | $45.2 \pm 2.0$ | $42.9 \pm 1.9$ | $41.4 \pm 1.6$ | $37.2 \pm 2.7$ | $30.5 \pm 1.3$ |
| WRI | $46.9 \pm 1.5$ | $45.4 \pm 1.9$ | $43.9 \pm 2.6$ | $39.5 \pm 3.2$ | $30.9 \pm 2.2$ |

to rely on color will be heavily penalized at test time. In this section, we describe the mechanism behind $\theta_{parity}$ in more detail.

We begin by sampling distribution parameters for all the invariant and conditional spurious datasets (the same as in prior experiments). Next, we select a random unit vector $\mathbf{c}_y \sim U(\mathbb{S}_{d_{spu}})$ for each class label. Our goal is to ensure that the spurious mean $\mu^{e,y}$ correlates positively with $\mathbf{c}_y$ for each of the training environments and negatively with the test environment. In other words, the training and test environments have opposite *parity* w.r.t. the hyperplane. We achieve this by negating $\mu^{e,y}$ when necessary, i.e.

$$\mu^{e,y} \leftarrow \begin{cases} \text{sign}(\mathbf{c}_y^{\mathsf{T}}\mu^{e,y})\mu^{e,y} & \text{if } e \in \mathcal{E}_{tr} \\ -\text{sign}(\mathbf{c}_y^{\mathsf{T}}\mu^{e,y})\mu^{e,y} & \text{otherwise} \end{cases}. \tag{C.10}$$

While this ensures the desired correlation, the relationship is not very strong due to the fact that random vectors in high-dimensional space tend to be nearly orthogonal. Therefore, we introduce $\theta_{parity}$.

For each class, we transform $\mu^{e,y}$ again to ensure it is inside the hypercone with axis $\mathbf{c}_y$ and opening angle $\theta_{parity}$. This is accomplished by rotating $\mu^{e,y}$ towards the axis so that the angle to the axis is scaled by a factor of $\theta_{parity}/90°$. The identity transform is, therefore, synonymous with $\theta_{parity} = 90°$, and as $\theta_{parity}$ decreases towards $0°$ the spurious data becomes more consistently predictive of the label.

**Numerical Results**   For experiments on the synthetic datasets, we integrate the DomainBed implementations of IRM and VREx. All experiments use 20 dimensional data (10 invariant dimensions and 10 spurious dimensions). They also all use 5 environments (1 test and 4 training) and 5 class labels. The numerical values visualized in the Figure C.2 bar graphs are shown in Tables C.1, C.2, and C.3.

### C.3 HETEROSKEDASTIC CMNIST

ColoredMNIST (CMNIST) is a dataset proposed by Arjovsky et al. (2019) as a binary classification extension of MNIST where the shapes of the digits are invariant features and the colors of the digits are spurious features that are more tightly correlated to the label than the shapes are, with the correlation between the color and the label being reversed in the test environment. Specifically, digits 0–4 and 5–9 are classes 0 and 1 respectively; after injecting 25% label noise to all environments, the red digits in the first two environments have an 80% and 90% chance of belonging to class 0, while the red digits in the third environment have a 10% chance. The design of the dataset allows invariant predictors (that only base their predictions on the shape of the digits) to outperform predictors that use spurious color information, ideally achieving 75% accuracy on all environments.

**Heteroskedastic CMNIST with Covariate Shift**  Variants of CMNIST have been proposed, with the aim of incorporating additional forms of distribution shift (Ahuja et al., 2020; Wu et al., 2020; Krueger et al., 2021). To demonstrate the efficacy of our method under heteroskedastic covariate shift, we construct a heteroskedastic variant of CMNIST, which we call HCMNIST, where label flips occur for digits 0, 1, 5, and 6 with probability 5% and label flips occur for other digits with probability 25%. Additionally, we generate a variant of HCMNIST with covariate shift, HCMNIST-CS, where we redistribute data among environments. We place 65% of the digits 0, 1, 5, and 6 into the first training environment and 5% into the second environment (with the remaining 30% in test). The remaining digits are distributed so that all environments have the same number of samples.

The empirical results on the HCMNIST with and without covariate shift are shown in Table 1. These experiments use the practical version of the WRI algorithm and are computed using the same evaluation strategy as DomainBed. The results demonstrate that the practical WRI implementation performs well in both the heterogeneous case with and without covariate shift. Conversely, both IRM and VREx have significant degradation when covariate shift is introduced.

We also create an idealized version of these datasets that simplifies the images into two-dimensional features consisting of the digit value and the color. We evaluate the WRI penalty and VREx penalty using three optimal predictors: one that operates on only the digit value (invariant), one that operates on the color value (spurious), and one that operates on both. The results of this experiment are presented in Table 2. As expected, we find that both WRI and VREx have zero penalty on HCMNIST. However, in the presence of covariate shift, the VREx penalty for the invariant predictor is greater than both the penalty for the spurious classifier and the penalty for the mixed classifier. This demonstrates a simple and concrete case where VREx does not select for an invariant classifier, suggesting that the degradation in VREx performance under shift in Table 1 can be attributed to a true failure of VREx to recover the invariant predictor.

### C.4 OUT-OF-DISTRIBUTION DETECTION WITH CMNIST

To evaluate the out-of-distribution (OOD) detection performance on CMNIST, we first train a model on the two training environments of (traditional) CMNIST, then assess the model's ability to classify sample digits as in-domain or out-of-domain on a modified version of the CMNIST test environment. The modified test environment is a mix of unaltered digits for the "in-domain" samples and flipped digits for the "out-of-domain" samples. Specifically, we horizontally flip digits 3, 4, 7, and 9, and vertically flip digits 4, 6, 7, and 9. In this way, we create CMNIST samples that would not exist in the training distribution but still appear plausible. Figure C.3 provides examples of these in-domain and out-of-domain samples. Note that we only flip digits that are unlikely to be confused with real digits after flipping.

In order to measure OOD performance, we require an OOD score for each sample that predicts if the sample is in-distribution. Each OOD score is then compared against its in-distribution label to compute a Receiver Operating Characteristic (ROC) curve. This curve captures the trade-off between the true positive rate and false positive rate at various confidence thresholds. Lacking an explicit density estimate for ERM, IRM, and VREx, we instead use the maximum class prediction score for the OOD score. We then compare the effectiveness of WRI's density estimate as an OOD score against ERM, IRM, and VREx. Finally, we calculate the area under the ROC curve (AUROC) as an aggregate metric. The results, presented in Table 3, indicate that the WRI density estimates more accurately detect OOD digits than the prediction confidence scores from the other methods.

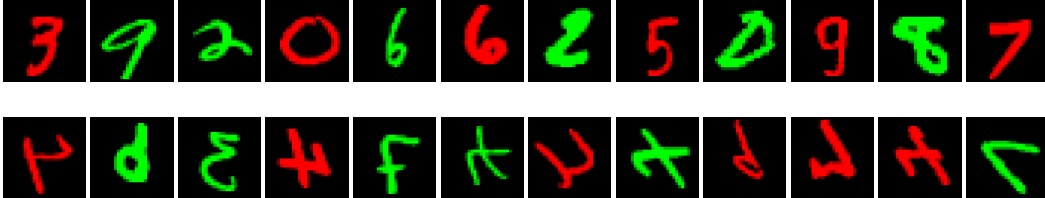

**Figure C.3:** Examples of CMNIST in-distribution (top) and out-of-distribution (bottom) used for OOD detection. The out-of-distibution samples are generated by flipping a subset of digits horizontally or vertically.

## C.5 DomainBed experimental details and results

### C.5.1 Featurizer

We use a ResNet50 model to train featurizers using the default ERM DomainBed parameters for each dataset. We train one featurizer corresponding to each test environment with both 32 and 64 dimensional features. Each featurizer is trained on the training environments for a fixed number of steps. Once trained, the features are pre-computed and all experiments are then performed on these features. The dimensionality of the features used is selected randomly as a hyperparameter. For each experiment, we use a single hidden-layer MLP with ReLU activation which operates directly on ResNet features.

### C.5.2 Hyperparameters

Since we are training on pre-extracted features, running for 5000 steps is unnecessary. Instead, we reduce the total number of steps per experiment to 500 and increase the default learning rate to 1e-2. Because we have fewer steps than the original (non-featurized) DomainBed implementation, we reduce the annealing iterations on IRM and VREx by a factor of 10 and limit the maximum annealing steps to 500. Otherwise, VREx and IRM would nearly always finish training before annealing has finished.

We specified the following DomainBed hyperparameters and selected them according to the following distributions. Note that in DomainBed, the default hyperparameters are used during the first hyperparameter seed, and random hyperparameters are selected for subsequent seeds. For any hyperparameters not listed here, we use the defaults provided by DomainBed.

- Learning-rate - default: 1e-2, random: log-uniform over $[5e\text{-}3, 1e\text{-}1]$.
- IRM-anneal iters - default $50$, random: log-uniform over $[1, 500]$. Number of steps before penalty weight is increased.
- VREx-anneal iters - default $50$, random: log-uniform over $[1, 500]$. Number of steps before penalty weight is increased.
- Featurizer dimensions - default: $64$, random: discrete uniform from $\{32, 64\}$. The dimensionality of the pretrained features used.
- WRI-$\lambda$ - default: 1, random: log-uniform over $[1e\text{-}1, 5e1]$. The penalty weight used when computing $\mathcal{L}$ in the predictor optimization step.
- WRI-annealing - default: $0$, random: discrete uniform from $\{0, 10\}$. Number of steps before WRI regularization term is included in loss function.
- WRI-density update freq ($\omega$) - default: $1$, random: discrete uniform from $\{1, 2, 4, 8\}$. Number of predictor optimization steps between density optimization steps.
- WRI-density learning rate - default: 2e-2, random: log-uniform over $[1e\text{-}2, 5e\text{-}2]$. Learning rate used for the density estimate optimizer.
- WRI-density weight decay - default: 1e-5, random: log-uniform over $[1e\text{-}6, 1e\text{-}2]$. The weight decay used for the density estimate optimizer.
- WRI-density batch size - default: $256$, random: discrete uniform from $\{128, 256\}$. The batch size used when optimizing for density estimates.

- WRI-density $\lambda$ - default: 5, random: log-uniform from $[5, 50]$. The penalty weight used when computing $\mathcal{L}$ in the density optimization step.

- WRI-density $\beta$ - default: 2e2, random: log-uniform from $[5e\text{-}2, 5]$. The negative log penalty weight used when computing $\mathcal{L}$ in the density optimization step.

- WRI-density steps $(n_d)$ - default: 4, random: discrete uniform from $\{4, 16, 32\}$. The number of optimization steps taken each time the density estimators are updated.

- WRI-min density - default: 0.05, random: uniform from $[0.01, 0.2]$. The minimum density imposed via scaled shifted sigmoid activation on density estimator models.

- WRI-max density - default: 1, random: uniform from $[0.4, 2]$. The maxmimum density imposed via scaled shifted sigmoid activation on density estimator models.

### C.5.3 DOMAINBED WITH ADDITIONAL BASELINES

We run DomainBed experiments on additional baselines to place our work in larger context, expanding Table 4 to Table C.4. Specifically, we also compare to GroupDRO (Sagawa et al., 2019), Mixup (Zhang et al., 2017), MLDG (Li et al., 2018), and CORAL (Sun & Saenko, 2016). These are the (current) top-performing methods that are implemented and tested in the DomainBed test suite that can be used with an ERM-trained featurizer—as that is sufficient to learn the invariant features necessary to train an OOD predictor (Rosenfeld et al., 2022), but also significantly speeds up training. We did not include methods that are not implemented in DomainBed, but it is worth mentioning examples like MIRO (Cha et al., 2022) that demonstrate state-of-the-art performance on image data. Note that all of the aforementioned methods are different lines of work; they are not causally motivated, but can sometimes have better generalization accuracy than methods with a causal basis.

**Table C.4:** DomainBed results on feature data, with additional non-causal baselines

| Algorithm | VLCS | PACS | OfficeHome | TerraIncognita | DomainNet | Avg |
|---|---|---|---|---|---|---|
| ERM | $76.5 \pm 0.2$ | $84.7 \pm 0.1$ | $64.5 \pm 0.1$ | $51.2 \pm 0.2$ | $33.5 \pm 0.1$ | 62.0 |
| IRM | $76.7 \pm 0.3$ | $84.7 \pm 0.3$ | $63.8 \pm 0.6$ | $52.8 \pm 0.3$ | $22.7 \pm 2.8$ | 60.1 |
| GroupDRO | $77.0 \pm 0.2$ | $84.8 \pm 0.1$ | $65.1 \pm 0.1$ | $51.3 \pm 0.2$ | $30.9 \pm 0.1$ | 61.9 |
| Mixup | $76.6 \pm 0.2$ | $85.0 \pm 0.1$ | $66.1 \pm 0.0$ | $52.3 \pm 0.7$ | $33.1 \pm 0.1$ | 62.6 |
| MLDG | $75.8 \pm 0.2$ | $82.3 \pm 0.2$ | $64.8 \pm 0.2$ | $48.3 \pm 0.2$ | $33.0 \pm 0.1$ | 60.8 |
| CORAL | $76.5 \pm 0.2$ | $85.1 \pm 0.2$ | $65.0 \pm 0.1$ | $51.8 \pm 0.4$ | $33.5 \pm 0.1$ | 62.5 |
| VREx | $76.7 \pm 0.2$ | $84.8 \pm 0.2$ | $64.6 \pm 0.2$ | $52.2 \pm 0.3$ | $26.6 \pm 2.1$ | 61.0 |
| WRI | $77.0 \pm 0.1$ | $85.2 \pm 0.1$ | $64.5 \pm 0.2$ | $52.7 \pm 0.3$ | $32.8 \pm 0.0$ | 62.5 |

### C.5.4 INDIVIDUAL DATASET RESULTS

This section contains the individual DomainBed results on VLCS, PACS, OfficeHome, TerraIncognita, and DomainNet. The Average column from each dataset table is reported in Table C.4. For all other dataset tables, the column labels indicates which environment was held out for test.

**VLCS**

| Algorithm | C | L | S | V | Avg |
|---|---|---|---|---|---|
| ERM | $97.2 \pm 0.1$ | $66.0 \pm 0.6$ | $68.9 \pm 0.5$ | $73.8 \pm 0.5$ | 76.5 |
| IRM | $97.0 \pm 0.0$ | $66.3 \pm 0.5$ | $69.1 \pm 0.5$ | $74.2 \pm 0.2$ | 76.7 |
| GroupDRO | $96.9 \pm 0.2$ | $67.1 \pm 0.4$ | $69.3 \pm 0.2$ | $74.6 \pm 0.3$ | 77.0 |
| Mixup | $97.3 \pm 0.1$ | $66.7 \pm 0.4$ | $68.7 \pm 0.5$ | $73.7 \pm 0.1$ | 76.6 |
| MLDG | $97.2 \pm 0.1$ | $63.9 \pm 0.6$ | $68.6 \pm 0.3$ | $73.7 \pm 0.3$ | 75.8 |
| CORAL | $96.9 \pm 0.2$ | $66.3 \pm 0.2$ | $68.8 \pm 0.5$ | $73.9 \pm 0.3$ | 76.5 |
| VREx | $97.2 \pm 0.1$ | $66.9 \pm 0.0$ | $68.4 \pm 0.5$ | $74.2 \pm 0.4$ | 76.7 |
| WRI | $97.1 \pm 0.2$ | $67.0 \pm 0.2$ | $69.6 \pm 0.6$ | $74.3 \pm 0.2$ | 77.0 |

**PACS**

| Algorithm | A | C | P | S | Avg |
|-----------|---|---|---|---|-----|
| ERM | $80.5 \pm 0.3$ | $81.5 \pm 0.2$ | $96.6 \pm 0.1$ | $80.2 \pm 0.1$ | 84.7 |
| IRM | $80.8 \pm 0.9$ | $82.0 \pm 0.3$ | $96.3 \pm 0.2$ | $79.9 \pm 0.3$ | 84.7 |
| GroupDRO | $81.4 \pm 0.3$ | $81.2 \pm 0.3$ | $96.4 \pm 0.2$ | $80.2 \pm 0.1$ | 84.8 |
| Mixup | $80.9 \pm 0.3$ | $81.7 \pm 0.3$ | $96.6 \pm 0.2$ | $80.8 \pm 0.2$ | 85.0 |
| MLDG | $80.0 \pm 0.5$ | $81.1 \pm 0.4$ | $95.7 \pm 0.2$ | $72.2 \pm 0.4$ | 82.3 |
| CORAL | $81.4 \pm 0.2$ | $81.8 \pm 0.4$ | $96.8 \pm 0.1$ | $80.5 \pm 0.2$ | 85.1 |
| VREx | $80.6 \pm 0.5$ | $81.8 \pm 0.7$ | $96.8 \pm 0.1$ | $80.0 \pm 0.2$ | 84.8 |
| WRI | $81.2 \pm 0.3$ | $82.3 \pm 0.1$ | $96.5 \pm 0.2$ | $80.7 \pm 0.1$ | 85.2 |

**OfficeHome**

| Algorithm | A | C | P | R | Avg |
|-----------|---|---|---|---|-----|
| ERM | $58.9 \pm 0.5$ | $50.2 \pm 0.3$ | $74.4 \pm 0.2$ | $74.5 \pm 0.4$ | 64.5 |
| IRM | $57.9 \pm 0.6$ | $49.8 \pm 0.5$ | $73.4 \pm 0.8$ | $74.1 \pm 0.8$ | 63.8 |
| GroupDRO | $59.7 \pm 0.3$ | $50.7 \pm 0.2$ | $74.8 \pm 0.1$ | $75.4 \pm 0.2$ | 65.1 |
| Mixup | $61.1 \pm 0.2$ | $52.2 \pm 0.1$ | $75.6 \pm 0.1$ | $75.7 \pm 0.1$ | 66.1 |
| MLDG | $59.1 \pm 0.4$ | $50.7 \pm 0.2$ | $74.8 \pm 0.1$ | $74.8 \pm 0.3$ | 64.8 |
| CORAL | $59.3 \pm 0.2$ | $50.6 \pm 0.4$ | $74.9 \pm 0.2$ | $75.2 \pm 0.2$ | 65.0 |
| VREx | $58.8 \pm 0.5$ | $50.4 \pm 0.3$ | $74.4 \pm 0.2$ | $74.8 \pm 0.4$ | 64.6 |
| WRI | $58.9 \pm 0.6$ | $49.8 \pm 0.2$ | $74.7 \pm 0.1$ | $74.7 \pm 0.3$ | 64.5 |

**TerraIncognita**

| Algorithm | L100 | L38 | L43 | L46 | Avg |
|-----------|------|-----|-----|-----|-----|
| ERM | $50.3 \pm 0.4$ | $50.0 \pm 0.6$ | $57.8 \pm 0.3$ | $46.6 \pm 0.4$ | 51.2 |
| IRM | $53.6 \pm 0.7$ | $53.2 \pm 0.8$ | $57.7 \pm 0.2$ | $46.6 \pm 0.8$ | 52.8 |
| GroupDRO | $50.4 \pm 0.5$ | $50.1 \pm 0.3$ | $57.4 \pm 0.3$ | $47.3 \pm 0.3$ | 51.3 |
| Mixup | $53.1 \pm 1.8$ | $52.4 \pm 0.9$ | $57.1 \pm 0.5$ | $46.6 \pm 0.3$ | 52.3 |
| MLDG | $44.7 \pm 0.6$ | $49.3 \pm 0.3$ | $57.2 \pm 0.2$ | $41.8 \pm 0.3$ | 48.3 |
| CORAL | $51.0 \pm 0.5$ | $51.5 \pm 1.0$ | $57.8 \pm 0.2$ | $47.0 \pm 0.5$ | 51.8 |
| VREx | $52.5 \pm 1.1$ | $51.2 \pm 0.6$ | $57.9 \pm 0.3$ | $47.2 \pm 0.4$ | 52.2 |
| WRI | $51.7 \pm 0.5$ | $55.0 \pm 0.6$ | $57.2 \pm 0.5$ | $47.1 \pm 0.3$ | 52.7 |

**DomainNet**

| Algorithm | clip | info | paint | quick | real | sketch | Avg |
|-----------|------|------|-------|-------|------|--------|-----|
| ERM | $47.9 \pm 0.2$ | $16.3 \pm 0.1$ | $40.6 \pm 0.2$ | $9.6 \pm 0.2$ | $46.7 \pm 0.2$ | $40.2 \pm 0.1$ | 33.5 |
| IRM | $31.9 \pm 4.4$ | $11.4 \pm 1.3$ | $28.9 \pm 3.0$ | $6.5 \pm 0.7$ | $30.8 \pm 3.9$ | $26.9 \pm 3.7$ | 22.7 |
| GroupDRO | $43.8 \pm 0.3$ | $15.8 \pm 0.2$ | $37.5 \pm 0.3$ | $8.8 \pm 0.1$ | $42.6 \pm 0.2$ | $37.1 \pm 0.3$ | 30.9 |
| Mixup | $47.2 \pm 0.1$ | $16.3 \pm 0.2$ | $40.2 \pm 0.2$ | $9.7 \pm 0.1$ | $45.5 \pm 0.3$ | $39.6 \pm 0.2$ | 33.1 |
| MLDG | $46.9 \pm 0.2$ | $16.2 \pm 0.1$ | $40.0 \pm 0.2$ | $9.2 \pm 0.1$ | $46.2 \pm 0.1$ | $39.4 \pm 0.1$ | 33.0 |
| CORAL | $47.9 \pm 0.3$ | $16.5 \pm 0.1$ | $40.5 \pm 0.2$ | $9.6 \pm 0.1$ | $46.6 \pm 0.2$ | $39.8 \pm 0.2$ | 33.5 |
| VREx | $37.1 \pm 3.2$ | $14.0 \pm 0.9$ | $31.9 \pm 2.8$ | $7.5 \pm 0.7$ | $37.2 \pm 2.5$ | $32.0 \pm 2.5$ | 26.6 |
| WRI | $46.7 \pm 0.2$ | $15.9 \pm 0.1$ | $39.8 \pm 0.1$ | $9.3 \pm 0.1$ | $46.0 \pm 0.2$ | $39.4 \pm 0.1$ | 32.8 |

