# OpenReview forum: "Weighted Risk Invariance for Density-Aware Domain Generalization"
_ICLR.cc/2024/Conference — Submitted to ICLR 2024_

### Official Review · Reviewer_nCje · 2023-10-31

**Soundness:** 2 fair
**Presentation:** 2 fair
**Contribution:** 2 fair
**Rating:** 5
**Confidence:** 3

**Summary:**

This paper proposes a novel domain generalization method, Weighted Risk Invariance (WRI) to learn invariant features across different domains. By considering a linear causal setting, given several assumptions, the authors theoretically show that WRI provides an invariant predictor. They next introduce an empirical algorithm WRI, in which the model parameters and the density of the invariant feature distribution are jointly learned in an alternating minimization scheme. Experimental results on both synthetic (ColoredMNIST) and real-world (DomainBed) data show that WRI provides better performance in both classification and out-of-distribution (OOD) detection compared to two relevant baselines.

**Strengths:**

The paper is well-written and easy to follow. The idea of introducing a Weighted Risk Invariance (WRI) approach to learn invariant features in domain generalization is interesting and novel. I appreciate the authors' effort to provide a theoretical guarantee of the satisfaction of the weighted risk invariance of their method. In addition, the empirical results of the paper demonstrate the benefits of the proposed WRI compared to two relevant approaches.

**Weaknesses:**

While I am aware that developing a fully theoretically sound domain generalization method is currently a big challenge in the community, my major concern is the main technical contributions of the paper. In particular,

1. The key idea of the proposed relies on the assumption (depicted in a causal graph in Fig. 2) that the observed feature $X$ can be decomposed by $X_{inv}$ and $X_{spu}$ without any details or explicit explanations (in the method and also in the implementation of the algorithm) about the way to extract the invariant feature $X_{inv}$ from $X$. To obtain the key factor $X_{inv}$, one often has to apply a causal discovery algorithm [A], which is relatively complicated and time-consuming.

2.  The definition of an invariant predictor (in Defn. 1) is not well-defined. Indeed, the invariance of the conditional distribution $p_e(f(X)|X_{inv})$ is not equivalent to the condition $f(X)=f(X_{inv})$. Furthermore, the domain invariant presentation in a general domain generalization should be based on the conditional distribution of the label given the input feature, i.e., $p_e(Y|g(X))$ with $g$ is a presentation mapping, not the predictor $f$.

3. Though I did not have time to verify the proof of Proposition 1, I am still not convinced the result that the invariance of the predictor $f$ leads to the weighted invariance (Defn. 2). In addition, could you please clarify the real meaning of the definition of the weighted invariance (Defn. 2), also the way to verify that property?  I also would like to see the comments from other reviewers on this.

4.  The experimental results of the paper are supportive, but not very convincing. In particular, it seems that the proposed WRI can only beat the related baselines (IRM and VREx) on the synthetic dataset (MNIST), while on the real-world dataset Domainbed, their performance is much similar, and WRI's even could not beat the naive ERM approach.

[A]. @article{nogueira2022methods,
  title={Methods and tools for causal discovery and causal inference},
  author={Nogueira, Ana Rita and Pugnana, Andrea and Ruggieri, Salvatore and Pedreschi, Dino and Gama, Jo{\~a}o},
  journal={Wiley interdisciplinary reviews: data mining and knowledge discovery},
  volume={12},
  number={2},
  pages={e1449},
  year={2022},
  publisher={Wiley Online Library}
}

**Questions:**

Please see my comments/questions in the Weaknesses part above.

---

> ### Author Response · Authors · 2023-11-18
>
> We thank the reviewer for their thoughtful engagement. We are glad that our approach seems both interesting and novel, and we are grateful for the reviewer’s recognition of the work that went into the theoretical justification. We hope that by answering the reviewer’s concerns, we can better clarify our technical contributions.
>
> Weaknesses
> 1. We wish to clarify two points in this regard:
>
> 	**a**. Our theory shows that under the model we consider, WRI learns a model that only relies on $X_{inv}$. This is the required condition for OOD generalization in this case, and is the goal of our method and implementation.
>
> 	Our goal is not causal discovery, and hence we do not wish to prove that some representation obtained by the model corresponds explicitly to $X_{inv}$. This is in line with many other works on learning invariant representations, e.g. IRM and many more follow-up works [1-3]. This is different from works such ICP [4], that indeed recover causal features and therefore perform causal discovery.
>
> 	**b**. An important distinction between performing invariant representation learning and performing causal discovery, is that generally invariant features are not guaranteed to be causes of $Y$. This is also a distinction made in prior work, e.g. in Wald et al. [5] (Fig. 1, Fig. 2a and corresponding analysis). Our method can also be applied in the data generating processes studied there, where $X_{inv}$ is an anti-causal feature that is a stable predictor under distribution shifts.
>
> 	We added a clarification about this distinction between causal discovery and invariant prediction to our discussion section, with thanks for the valuable input.
>
> 2. We thank the reviewer for their insight on this important point. We corrected our definition of an invariant predictor: now, we more precisely define a predictor that only uses invariant features as *spurious-free*, and we clarified the connection to the standard definition of invariant predictor as mentioned by the reviewer. We also modified our statement of Proposition 1 accordingly.
>
> 3. We are confident in the accuracy of the claim and proof, and are open to answering any questions about the proof. We see how the claim may seem less intuitive due to the point regarding the meaning of the term invariance, which we discussed earlier in the response. However, we believe that using the term "spurious free", as we do in the updated paper, makes the claim much more intuitive (we emphasize that the claim is still mathematically correct according to the definitions we used in the earlier version as well, it is just the name of the mathematical property that we updated here).
>
> 	To explain why the claim holds intuitively, please note that if we write down the expected risk of a classifier that is only a function of the invariant features, $f(X_{inv})$, then the spurious features can be marginalized out and the risk is just the expected risk under the marginal distribution of $X_{inv}$ and $Y$. Because reweighting is done *only on the invariant features*, then the marginal distribution over $X_{inv}$ does not shift between environments and then the risk of any classifier that only depends on $X_{inv}$ is fixed across them. We hope that this clarifies the proof.
>
> 	The intuition behind the weighted risk in Definition 2 is to adjust samples so that environments with different distributions of invariant features can be compared. This is a similar concept to importance reweighting [6], but with adjustments to favor invariant learning. The weighted risk between two environments is asymmetric, and we use the phrase “weighed risk invariance” to describe when the risk of environment A, weighted by the density from environment B, is equal to the risk of environment B weighted by the density from environment A.
>
> 	We can verify that weighted risk invariance holds using the predictor and density functions we learn after training with eq 9. For an environment A, the predictor is used to compute the loss at each point, and the density function trained for another environment B is used to weight the losses. We then take the average over all the weighted losses at each point to approximate the risk of environment A weighted by the density from environment B. We flip the environments to compute the risk of environment B weighted by the density from A, and these quantities should be equal.
>
> [1] Jiang and Veitch (2022). Invariant and Transportable Representations.
>
> [2] Mitrovic et al (2021). Representation Learning via Invariant Causal Mechanisms
>
> [3] Rojas-Carulla et al (2018). Invariant Models for Causal Transfer Learning
>
> [4] Peters et al (2015). Causal inference using invariant prediction
>
> [5] Wald et al (2021). On calibration and out-of-domain generalization
>
> [6] Shimodaira (2000). Improving predictive inference under covariate shift.

---

> > ### Comment · Reviewer_nCje · 2023-11-22
> > **Thanks for the responses**
> >
> > I would like to thank the authors for their detailed responses, especially for the clarification of the experiments on Domainbed. However, it is still my concern about how to find the term $X_{inv}$ without using causal discovery. Maybe the authors should not mention "causal inference" in the problem setting to avoid misunderstanding. Therefore, I decide to retain the original score for the paper.

---

> > > ### Author Response · Authors · 2023-11-22
> > >
> > > Please note that **the term "causal inference" does not appear anywhere in the manuscript**, our problem is defined as OOD generalization. Our work is causally-motivated, like many of the works cited previously [1-3]; like these other works, a causal model is important to motivating our assumptions and our data-generating process. However, we do not claim to be doing causal discovery.
> > >
> > > We kindly ask that the reviewer consider that we have a different problem definition, one that is in line with extensive work on causally-motivated methods.
> > >
> > > [1] Jiang and Veitch (2022). Invariant and Transportable Representations.
> > >
> > > [2] Mitrovic et al (2021). Representation Learning via Invariant Causal Mechanisms
> > >
> > > [3] Rojas-Carulla et al (2018). Invariant Models for Causal Transfer Learning

---

> ### Author Response · Authors · 2023-11-18
>
> Weaknesses (cont'd)
>
> 4. Before addressing this concern, we would first like to clarify some of our results on the DomainBed test suite. DomainBed comprises 5 real world datasets: VLCS, PACS, OfficeHome, TerraIncognita, and DomainNet. These datasets all include a variety of image styles and natural shifts. Generalizability is not only measured by performance on individual datasets, but also on the datasets on average. Over all the datasets, WRI performs the best on average, and also outperforms ERM on all datasets but OfficeHome (where it has the same average accuracy) and DomainNet. In all cases but DomainNet, WRI is also the clear first by outperforming the other methods by more than a standard deviation, or else it is within a standard deviation of the best method. Except for ERM on DomainNet, this cannot be said for any other method/dataset combination.
>
> 	That being said, we do recognize the reviewer’s concerns about the similar performance levels. We would like to emphasize that in addition to achieving competitive prediction accuracy, our method is unique in that it also trains density estimators for invariant features, and these allow users to get additional information at test time. We demonstrate empirically that this density is useful for OOD detection, but it should be useful for detecting rare/minority examples in general. This is an additional signal that other invariant prediction methods do not provide.

---

> ### Author Response · Authors · 2023-11-21
>
> Thank you for taking the time to read and review our submission. We would be happy to answer any remaining questions before the discussion period ends tomorrow.

---

### Official Review · Reviewer_G5hc · 2023-10-31

**Soundness:** 2 fair
**Presentation:** 2 fair
**Contribution:** 2 fair
**Rating:** 3
**Confidence:** 3

**Summary:**

In this paper, the authors introduce a novel approach to deal with out-of-distribution generalization, particularly when invariant features are subjected to covariate shift. The method emphasizes the utilization of the weighted risk between diverse environments to ensure an invariant predictor. The authors provide theoretical guarantees for the identification of invariant features in linear data-generation processes. Empirical results on various datasets demonstrate the effectiveness of the proposed methodology.

**Strengths:**

1. Addressing out-of-distribution generalization is of paramount importance, given its wide applications.
2. The invariance regularizer, grounded in weighted risks, is both theoretically sound and interesting.
3. The paper is well-organized.

**Weaknesses:**

1. The authors assert that REx is limited to the homoskedastic setting, whereas their method can accommodate the heteroskedastic setting. However, the definitions appear to pertain to disparate scenarios. In the REx literature, homoskedasticity is tied to noise variance discrepancies across different $X$, while heteroskedasticity in this work relates to covariate shifts in invariant features. The rationale behind REx's inability to address the heteroskedastic covariate shift is not lucid. Moreover, the proposed WRI seems incapable of addressing the conventional heteroskedastic scenario, as varying noise variance for $Y$ across environments would render the weighted risk inconsistent across environments.
2. The "Comparison to IRM" section is unclear. A formal proof delineating the superiority of WRI over IRM would enhance clarity. Integrating the REx loss in Figure 4 might also be beneficial.
3. The implementation details are somewhat unclear. The reason behind employing an alternating minimization process, as opposed to direct optimization of Equation (9), is not explicit. Furthermore, ensuring the identification of invariant features via Equation (9) seems challenging. Notably, the final term in Equation (9) gravitates towards $d(x) \rightarrow P_e(x)$, inclining towards a dependency on all features. Furthermore, it is unclear whether the second term of Equation (9) also contains other trivial solutions (like the all zero solution mentioned by the author).
4. The omission of critical baselines, such as SWAD [1] and MIRA [2], potentially diminishes the empirical significance of the proposed technique. The authors' assertion in the appendix that non-causally motivated methods can occasionally outperform causally-based methods in domain generalization tasks appears to undermine the essence of leveraging causality-based techniques in this realm.
5. In the "OOD detection performance of our learned densities" section, there is an absence of detailed explanation regarding the modified CMNIST test split incorporating mirrored digits.
6. The discussion would benefit from a more extensive review of related work addressing covariate-shift generalization, e.g., [3][4].


[1] Cha, Junbum, et al. "Swad: Domain generalization by seeking flat minima." Advances in Neural Information Processing Systems 34 (2021): 22405-22418.

[2] Arpit, Devansh, et al. "Ensemble of averages: Improving model selection and boosting performance in domain generalization." Advances in Neural Information Processing Systems 35 (2022): 8265-8277.

[3] Xu, Renzhe, et al. "A theoretical analysis on independence-driven importance weighting for covariate-shift generalization." International Conference on Machine Learning. PMLR, 2022.

[4] Duchi, John, Tatsunori Hashimoto, and Hongseok Namkoong. "Distributionally robust losses for latent covariate mixtures." Operations Research 71.2 (2023): 649-664.

**Questions:**

See the weakness part.

---

> ### Author Response · Authors · 2023-11-18
>
> We thank the reviewer for their thoughtful engagement, and we are glad that the reviewer finds our approach theoretically sound and interesting. However, we feel there may be a misunderstanding of some of our contributions.
>
> Weaknesses
>
> 1. To answer this comment, we wish to lay out the following points:
>
> 	**a**. Heteroskedasticity in our setting is exactly the same as defined in REx. Under heteroskedasticity, different values of $X$ have different levels of noise. We see that perhaps our use of the phrase "heteroskedastic covariate shift" (in paragraph 4 of the Introduction and the Conclusion) may be misleading in that sense, and we changed that phrasing to clarify that our meaning is exactly the same as in REx.
>
> 	**b**. Covariate shift is a completely distinct notion from heteroskedasticity. However, our point is that under heteroskedasticity, covariate shift leads the invariant classifier to have different risks across environments. Thus, REx will not learn a correct classifier.
>
> 	**c**. The above point is recognized in the REx paper, where the authors give the following explanation:
>
> 	> REx…performs poorly in the domain-heteroskedastic case….Intuitively, this is because the irreducible error varies across domains in these tasks, meaning that the risk will be larger on some domains than others.
>
> 	We offer this explanation via example in “Comparison to REx”. Following your suggestion, we added details to make the example more explicit.
>
> 	**d**. We would appreciate further clarification on what you mean by varying noise variance for $Y$ across environments. If $Y$'s variance changes across environments, then $p_e(Y \mid X_{inv})$ is not consistent across environments, hence the overwhelming majority of invariant learning algorithms should not learn a classifier proportional to $p_e(Y \mid X_{inv})$. Note that this setting is different than heteroskedastic noise, where noise levels may change across $X$ but are fixed across environments (i.e. $p_e(Y \mid X_{inv})$ is the same for all $e$, but $Y$ may have varying noise levels across values of $X_{inv}$).
>
> 2. In “Comparison to IRM”, we show that as the degree of covariate shift between two environments increases, then the IRM penalty becomes less sensitive to invariance violations, resulting in increased sample complexity. The sample complexity issues of IRM are already extensively discussed in previous work, often where the entire paper is focused on analyzing the failure cases of IRM [1-2]. The focus of our paper is on the potential benefit of invariance weighting in invariant learning, so another proof of the sample complexity of IRM would be out of scope.
>
> 	We thank the reviewer for the recommendation for Figure 4, but we suggest that adding REx to Figure 4 may be misleading. Depending on how difficult examples are distributed across the two environments, the REx penalty may be misaligned with recovering an invariant predictor (for reasons discussed above). Thus, the magnitude of the REx penalty may not even be relevant to its sensitivity to invariance violations.
>
> 3. The reason we employ alternating minimization is precisely due to the difficulty of identifying the invariant features. Because we do not have access to these, we cannot learn their density by directly optimizing equation 9. Instead, we rely on the observation that if $d(X) = p_e(X_{inv})$, then an invariant predictor would have zero WRI penalty, and if we have invariant features, then we can perform density estimation on those features. With the density regularization term, we find that we do not recover any other trivial solutions in practice.
>
> 4. Causally-motivated methods have the benefit of interpretability and sound theoretical grounding. As a trade off, they can be difficult to scale to complex, high dimensional data, and therefore cannot always achieve the generalization accuracy of non-causal methods. We do reference and compare to several of these other methods to place our work in context (in the section of the Appendix you mention). Overall, we believe that pushing forward the state of the art in causally-motivated methods is, in itself, a significant contribution.
>
> 5. To address this, we created Appendix section C.4 to include a more detailed explanation of the modified CMNIST test split; this should better clarify how we create examples that are OOD in the invariant feature space. We thank the reviewer for the suggestion.
>
> 6. We thank the reviewer for the references, which we have added to the discussion section. We do want to emphasize that these works are about general covariate shift, and do not discuss the particular setting we are dealing with here, which is covariate shift over latent invariant features.
>
> [1] Ahuja et al (2021). Empirical or Invariant Risk Minimization?
>
> [2] Rosenfeld et al (2021). Risks of Invariant Risk Minimization

---

> ### Author Response · Authors · 2023-11-21
>
> Thank you for taking the time to read and review our submission. We would be happy to answer any remaining questions before the discussion period ends tomorrow.

---

> > ### Comment · Reviewer_G5hc · 2023-11-22
> >
> > I appreciate the authors' response and would like to delve deeper into a couple of points for further clarification.
> >
> > Regarding Point (1)d, I have revisited the example provided in the context of IRM where $X_1 \sim N(0, \sigma^2)$, $Y = X_1 + N(0, \sigma^2)$, and $X_2 = Y + N(0, 1)$. In this scenario, $X_1$ is identified as the invariant feature, and the environments are characterized by different values of $\sigma^2$. Consequently, the probability $P(Y|X_1)$ varies across these environments. Given this variability, I am concerned about the validity of the claim that the majority of invariant learning algorithms should not learn a classifier proportional to $P(T | X_{inv})$.
> >
> > Additionally, concerning Point 3, I still hold the view that the methods section requires more detailed explanation, particularly in terms of how the proposed alternative training techniques effectively identify invariant features.

---

> ### Author Response · Authors · 2023-11-22
>
> **Regarding how IRMv1 and other invariant learning methods recover $P(Y \mid X_{inv})$.** In the specific case of IRMv1, they define invariance as $\mathbb{E}\_{p_e}[Y \mid \phi(X)] = \mathbb{E}\_{p_{e'}}[Y \mid \phi(X)]$ for any two environments $p_e$ and $p_{e'}$, instead of the full conditional probabilities ($p_e[Y \mid \phi(X)] = p_{e'}[Y \mid \phi(X)]$). Indeed in their model the conditional mean is invariant, yet as you suggested, the conditional distributions aren't.
> Since minimizing the squared loss only involves finding the conditional mean, their analysis works out in that case and it is guaranteed to learn a predictor that is free of spurious correlations.
>
> Hence, they just use a weaker form of conditional independence, whereas subsequent work discusses many forms of such conditional "independencies", showing cases where these are appropriate. For instance, [1] also match the variance instead of the mean, while [2] and many others seek to match the full conditional probabilities (i.e. having $p_e[Y \mid \phi(X)] = p_{e'}[Y \mid \phi(X)]$). Invariance of the loss, which is what we study, is discussed in many other papers that are cited in our work, and conditions are drawn for cases where this is a sufficient condition to discard spurious correlations. In our model, like many other works such as ICP, REx, and others cited in our paper, we do assume that the noise level is constant across environments, and that $p(Y \mid X_{inv})$ is fixed across environments. We should also note that we choose to focus on risk invariance since it is a popular type of invariance to study, and the theory and main ideas in our paper can likely be applied with other invariance principles.
>
> **Regarding the discussion on how our objective and others recover invariant features.** We agree that this can be better clarified, and we have modified our implementation discussion (Appendix B) to give a more detailed discussion on this. The easiest way to explain this is perhaps starting from a constrained optimization problem, where weighted risk invariance is enforced as a constraint, and the objective is to minimize the training loss. Then the theoretical statement says that models with zero weights on spurious features will be the ones satisfying the constraint. Optimizing the loss chooses the best performing one among those, which under the conditions of the model is just the optimal invariant predictor (corresponding to the conditional mean when we consider the squared loss or logistic loss for classification---note that trivial solutions that satisfy the constraint but do not minimize empirical loss will not be selected). Then our objective is simply a penalized version of this constrained problem. This largely follows a similar path to works like IRM and many of its follow-ups (and more generally works that approximate constrained problems with a penalty), hence we chose to focus on the properties of WRI in our submission as this type of development is not a contribution of our work. However, we see how in the process we lost some of the accessibility of our arguments to audiences that are not familiar with exactly this type of argumentation. Hence we have added the aforementioned discussion to our appendix and we will emphasize it in the main paper. Thanks very much for your engagement and your help in improving our paper.
>
> [1] Bellot, Alexis, and Mihaela van der Schaar. (2020) "Accounting for unobserved confounding in domain generalization." arXiv preprint arXiv:2007.10653.
>
> [2] Puli, Aahlad Manas, et al. (2021) "Out-of-distribution Generalization in the Presence of Nuisance-Induced Spurious Correlations." International Conference on Learning Representations.

---

### Official Review · Reviewer_924Z · 2023-11-01

**Soundness:** 3 good
**Presentation:** 3 good
**Contribution:** 3 good
**Rating:** 6
**Confidence:** 3

**Summary:**

The paper proposed Weighted Risk Invariance (WRI), a new optimization formulation for the OOD generalization problem. Particularly, the paper claims WRI to be able to recover invariant predictor even in the case of heteroskedastic covariate shift. The claim is justified theoretically under a linear causal setting. The paper also proposes a practical algorithm to solve modified WRI in the practical regime by alternating between learning the model parameters and the density of the invariant distribution. The experiments show outperforming results compared to other causally motivated methods.

**Strengths:**

- The paper is generally well written and easy to follow.
- The proposal is well formulated, novel, and non-trivial.
- Essential claims are theoretically justified for the linear causal setting.
- Experimental settings are detailed.

**Weaknesses:**

Although the paper is interesting, I am having the following issue/concerns:
- The introduction and comparison to (V)REx and IRM are separated, posing some difficulty in reading.
- The usage of density estimates in OOD detection are not detailed but only a brief description in section 4.
- No argument is provided for the non-linear case, even though the experiments on DomainBed does not follow the linear regime.
- Lacking analysis on the difference between WRI and the practical objective.

**Questions:**

My main concerns are the final two points above, since without addressing them, the experiments are divorced from the first half. I am happy to adjust my score if the author can address these concerns.

---

> ### Author Response · Authors · 2023-11-18
>
> We thank the reviewer for their thoughtful engagement. We are glad that the reviewer finds the proposal well formulated, novel, and non-trivial.
>
> Weaknesses
> 1. We combined the introduction and comparison to VREx and IRM in our latest version. We appreciate the suggestion to improve the readability of our paper.
> 2. The use of density estimates is a key benefit of our work. Following your suggestion, we added Appendix C.4 to include more technical detail. We now spend more time discussing how we construct our OOD samples, as well as how our density estimates are used as an OOD score (in line with current OOD literature [1]).
> 3. We agree with the reviewer that theory for the nonlinear case would be ideal. Our current theory revolves around systems of quadratic equations having unique solutions, and a nonlinear generalization would result in a different system of nonlinear equations that seems quite difficult to analyze and is beyond the scope of this paper. We wish to emphasize that existing literature only provides theoretical justification for either the linear case [2-4] or for highly specific nonlinear distributions under additional assumptions [5]. Many works in the area also do not provide theoretical justification at all. Therefore, we believe that a generalized theory for nonlinear models would be a truly seminal contribution which deserves its own work. In this work, our focus is on the effects of covariate shift in the invariant features, which we study both empirically and theoretically within the framework that is currently common in the literature on invariant learning.
>
>     In context with current literature, we believe our formulation is still a significant contribution. We first provide theoretical and empirical evidence for the linear model, then empirical evidence for the nonlinear model where some version of our causal structure holds (CMNIST), and finally empirical evidence on DomainBed, a benchmark of highly unstructured datasets with unknown shifts. In this way, we hope to form a coherent narrative from our theory to our experiments.
> 4. The practical objective aligns closely with the WRI objective: both use an ERM term (to encourage good performance across environments) and a WRI term (to encourage that we learn to predict on only the invariant features). The main difference between the two is in access to the density of the invariant features, so the practical objective additionally includes density as a learnable parameter and a regularization term for density estimation.
> Following your concern, we updated the paper to make the similarity/difference more clear (below equation 9).
>
> [1] Liu et al (2021). Energy-based OOD Detection
>
> [2] Peters et al (2015). Causal inference using invariant prediction
>
> [3] Arjovsky et al (2019). Invariant Risk Minimization
>
> [4] Krueger et al (2021). OOD Generalization via Risk Extrapolation
>
> [5] Dong and Ma (2023). First Steps Toward Understanding the Extrapolation of Nonlinear Models to Unseen Domains

---

> ### Author Response · Authors · 2023-11-21
>
> Thank you for taking the time to read and review our submission. We would be happy to answer any remaining questions before the discussion period ends tomorrow.

---

> > ### Comment · Reviewer_924Z · 2023-11-23
> >
> > I appreciate the authors' response and I am convinced to raise the score to 6. One additional suggestion after re-reading is that the authors should try put Appendix B as part of the main paper (Section 3.2) and highlight it as a key novelty.

---

> > > ### Author Response · Authors · 2023-11-23
> > >
> > > Thank you very much for your support! We will follow your suggestion to integrate Appendix B to the main paper. We appreciate your time and feedback, and for your help in improving our paper.

---

### Official Review · Reviewer_qV1P · 2023-11-03

**Soundness:** 3 good
**Presentation:** 4 excellent
**Contribution:** 3 good
**Rating:** 6
**Confidence:** 4

**Summary:**

This paper proposes a novel problem formulation called “Weighted Risk Invariance” for domain-invariant feature learning in domain generalization. Under the given causal model and a given a set of input environments, the goal of domain-invariant feature learning is to recover a domain invariant predictor, which ensures that the model is robust to distribution shifts in new environments by only relying on invariant features. However, this can be challenging when covariate shift occurs in the invariant features themselves - which requires accounting for the distributions of the potentially shifted invariant feature. This is achieved via the proposed notion of “Weighted” risk invariance, which can be solved by reweighting the ERM loss function with the marginal density of the invariant features. Further, a practical objective is proposed to ensure that this objective recovers non trivial solutions,  enforced via a negative log penalty term which discourages small density estimates. These claims are thoroughly supported with experiments on both synthetic and real life datasets - achieving competitive performance with previous baselines such as IRM.

**Strengths:**

Overall, the paper is well written, well motivated and easy to follow. More specifically:
1. The paper studies an important issue under a causal model where the invariant features themselves can shift across the observed train (and future test) environments. The proposed formulation of weighted risk invariance seems novel, and is more general / flexible setting to study domain invariance, going beyond the typically studied causal models.
2. The paper is clear and concise - the problem is motivated very well via appropriate illustrations which make it easier for the reader to understand the importance of this work. For example, the comparison with IRM on Page 6 is interesting. Similarly, the experiments on the new proposed MNIST versions is also an interesting setup.

**Weaknesses:**

Please see below:
1. While the paper does a good job of motivating the problem and showing results on synthetic setups, my main concern is reg. the experiments on real life datasets: that it is unclear whether a conclusion can be made. When does the proposed method work v/s when does it not, when it comes to real life datasets? Is there any reason behind these observations? What can one infer?
2. Following up on the previous point, could the authors explain when might such a phenomenon occur in a real life dataset more explicitly? Perhaps a real life intuition would help.

See more questions / suggestions in the next question.

**Questions:**

Questions:
1. Is there any assumption on the support of the invariant features? Is this why the practical objective has been proposed over the original formulation?
2. Is there any guidance for the reader on when this method should and should not be used, given a dataset?

Suggestions:
1. It might be a good idea to include a summary of all experimental results in a visual format e.g. plot so that the results are easier to read through.
2. It would be nice to visualize the new proposed colored MNIST example via an illustration to understand the setting used in the paper.

---

> ### Author Response · Authors · 2023-11-18
>
> We thank the reviewer for their engagement and support of our paper. We tried to study invariance in a more general setting, and we are glad that the reviewer finds our formulation novel and the results interesting.
>
> Weaknesses
>
> 1. Our main intention with the real life datasets was to demonstrate that our method does scale up to large image data, and is robust enough to perform well on a variety of image styles and shifts. However, current benchmarks like DomainBed are complex and difficult to parse in terms of what shift is occurring. Therefore, most proposed methods do not show a significant improvement on these benchmarks, and it is even unclear whether improvement on these specific benchmarks is the best way to evaluate generalization methods [1].
>
> 	Our goal is to raise an issue that we believe is relevant in any future application of invariant learning, and even though there are some more steps and further research required in order to make invariant learning relevant in practice, we hold that research into the common pitfalls of invariant learning methods is useful down the line as the literature forms more effective methods. Our application of reweighting to invariant learning may be incorporated as a component on top of future improvements in invariant learning methods.
>
> 2. Many real life scenarios exhibit heteroskedasticity and/or shift in the invariant features, settings our method aims to be robust to. As an example, we can imagine having data from various hospitals where demographics change. If we think of smoking as a causal feature for predicting cancer, and the percentage of smokers changes across environments, then this is a clear case of covariate shift in invariant features. Some types of cancer being more difficult to diagnose than others is a case of heteroskedasticity.
>
> Questions
>
> 1. We make no assumption on the support of the invariant features. We do want to point out that if there were no overlapping support in the invariant features at all, then there is no way to guarantee that our method, or any invariant prediction method, recovers an invariant classifier [2]; this is a fundamental limit of the invariant prediction problem. When there is some overlapping support, then invariant learning is possible.
>
> 	The practical objective has been proposed over the original formulation because the original formulation requires the density of the invariant features, and we do not usually have access to these in practice. Instead, we propose to learn both the invariant feature density and an invariant predictor via alternating minimization. We see that this results in our learning useful invariant feature densities (through our OOD experiments) in addition to our invariant predictor.
>
> 2. Good question. We believe whether our method should be used depends on the dataset and use case. To elaborate:
> 	- WRI should be considered when you are looking for a causally-motivated method. Causally-motivated methods can be seen as more interpretable and theoretically sound, although they can also be difficult to scale to high-dimensional data. If this is not important to your use case, we can recommend non-causal methods with strong empirical performance like MIRO [3] and Fishr [4].
> 	- In our paper, we compare to VREx as another causally-motivated baseline. When we have a setting with heteroskedasticity and shift in the invariant features (see our response to Weakness 2), then VREx will fail to recover an invariant predictor. Thus, we recommend WRI if you suspect your data to be more heterogeneous. Our performance should be similar to VREx otherwise—in this case, we would recommend just using VREx directly.
> 	- Learning the density estimates of invariant features is unique to our method. In an experiment, we demonstrated that this is useful for OOD detection. More generally, this is also useful for informing a user when they are encountering test samples that are rare/minority samples in training. Thus, we recommend our method if you want to learn an invariant predictor that also includes density information at test time—e.g. for fairness or bias mitigation reasons.
>
> [1]. Eastwood et al (2022). Probable Domain Generalization
>
> [2]. Ahuja et al (2021). Invariance Principle Meets Information Bottleneck for OOD Generalization
>
> [3]. Cha et al (2022). Domain generalization by mutual-information regularization
>
> [4]. Rame at al (2022). Fishr: Invariant gradient variances for OOD generalization

---

> ### Author Response · Authors · 2023-11-18
>
> Suggestions
>
> 1. Some of our experimental results are unconventional to present as a plot, e.g. CMNIST and DomainBed test results are always shown as tables, and these experiments do not have a continuously varying aspect that would benefit from being presented as a plot. However, we fully agree with the spirit of the suggestion; in trying to improve the readability of our results, we identified Table 2 as being difficult to read/interpret. Thus, we reworked Table 2 so that it matches the general format of the other tables (methods on the left, overall results on the right) and added some slight visuals. We believe this significantly improves its presentation and we thank the reviewer for the suggestion. Please feel free to raise additional concerns.
>
> 2. We appreciate the suggestion. We have created Appendix C.4 to visualize examples from the proposed CMNIST dataset and to include additional technical detail.

---

> ### Author Response · Authors · 2023-11-21
>
> Thank you for taking the time to read and review our submission. We would be happy to answer any remaining questions before the discussion period ends tomorrow.

---

### Author Response · Authors · 2023-11-18

We sincerely thank all the reviewers for their time and helpful feedback. Our paper addresses an important problem (reviewers qV1P, G5hc), and we are pleased that all the reviewers find our proposed formulation novel/interesting (all reviewers), well motivated (qV1P), and theoretically sound (924Z, G5hc). We are also pleased that our experiments are considered to be detailed and to demonstrate the benefit of our approach (924Z, nCje), as well as being easy to understand (qV1P).

Below, we summarize the changes we made to our revision in response to reviewer feedback:
- Modified our definition of an invariant predictor to specify when a predictor is *spurious-free* and when it is *optimally invariant/invariant* (Definition 1).
- Added discussion on our WRI objective and our practical objective (Appendix B)
- Added clarifying detail to our example of when REx fails to recover an invariant predictor (Section 3.1, “Comparison to REx”).
- Changed the description of the setting “heteroskedastic covariate shift” to “heteroskedasticity and covariate shift” for clarity (two instances).
- Added technical description and visualization for the OOD experiments (Appendix C.4).
- Reworked the format of Table 2 to incorporate visual elements and be more readable.
- Combined the introduction of VREx and IRM for readability (Section 3.1).
- Clarified the distinction between causal discovery and invariant prediction (Discussion).
- Included additional references on generalization under covariate shift for more complete context (Discussion).

We believe these modifications, added descriptions, and added figures helped improve the paper. We are very happy to engage with any remaining questions or concerns during the discussion period.

---

### Meta-Review · Area_Chair_1REa · 2023-12-08

**Metareview:**

This is borderline paper. After the authors' rebuttal, reviewers are still not convinced about some technical details and claims of the proposed method. Therefore, based on the current shape, this work is not ready for publication.

The authors are encouraged to further revise the paper to fully address reviewers' comments for a future submission.

**Justification For Why Not Higher Score:**

There are some technical concerns.

**Justification For Why Not Lower Score:**

N/A

---

### Decision · Program_Chairs · 2024-01-16

Reject